# VGLL1 cooperates with TEAD4 to control human trophectoderm lineage specification

Yueli Yang[1,11], Wenqi Jia[2,3,4,11], Zhiwei Luo[2,3], Yunpan Li[2,3], Hao Liu[2,3], Lixin Fu[4,5], Jinxiu Li[4,5], Yu Jiang[1], Junjian Lai[2,3,5], Haiwei Li[6], Babangida Jabir Saeed[2,3], Yi Zou[5], Yuan Lv [2,3,5], Liang Wu[2,3], Ting Zhou [7], Yongli Shan[3], Chuanyu Liu [5], Yiwei Lai[5,8], Longqi Liu [5,8,9], Andrew P. Hutchins [10], Miguel A. Esteban [1,2,3,5,8] ✉, Md. Abdul Mazid [2,3] ✉ & Wenjuan Li [2,3] ✉

In contrast to rodents, the mechanisms underlying human trophectoderm and early placenta specification are understudied due to ethical barriers and the scarcity of embryos. Recent reports have shown that human pluripotent stem cells (PSCs) can differentiate into trophectoderm (TE)-like cells (TELCs) and trophoblast stem cells (TSCs), offering a valuable in vitro model to study early placenta specification. Here, we demonstrate that the VGLL1 (vestigial-like family member 1), which is highly expressed during human and non-human primate TE specification in vivo but is negligibly expressed in mouse, is a critical regulator of cell fate determination and self-renewal in human TELCs and TSCs derived from naïve PSCs. Mechanistically, VGLL1 partners with the transcription factor TEAD4 (TEA domain transcription factor 4) to regulate chromatin accessibility at target gene loci through histone acetylation and acts in cooperation with GATA3 and TFAP2C. Our work is relevant to understand primate early embryogenesis and how it differs from other mammalian species.

The placenta is the largest fetal supporting organ. It originates from the trophectoderm (TE) and constitutes, together with the pluripotent inner cell mass (ICM), the first lineage bifurcation event during development[1]. The ICM gives rise to the hypoblast and the epiblast, which will form the yolk sac and the three embryonic germ layers, respectively. The TE gives rise to the cytotrophoblast and ultimately the placenta. The placenta mediates exchanges between the maternal uterus and the fetus by supplying nutrients, producing hormones, eliminating waste products, and enabling gas diffusion via maternal blood circulation. In line with these functions, abnormalities in placental development are one of the key reasons for early pregnancy loss and are also responsible for many pregnancy-related diseases including preeclampsia[2–4]. Due to ethical considerations and difficulties in obtaining study material, the genetic regulatory mechanisms underlying the formation of human TE and early cytotrophoblast remain unclear.

In contrast to human, TE development has been relatively well explored in the mouse. TEAD4 (TEA domain transcription factor 4) has

[1]State Key Laboratory for Diagnosis and Treatment of Severe Zoonotic Infectious Diseases, Key Laboratory for Zoonosis Research of the Ministry of Education, Institute of Zoonosis, and College of Veterinary Medicine, Jilin University, Changchun, China. [2]Laboratory of Integrative Biology, Guangzhou Institutes of Biomedicine and Health, Chinese Academy of Sciences (CAS), Guangzhou, China. [3]CAS Key Laboratory of Regenerative Biology and Guangdong Provincial Key Laboratory of Stem Cells and Regenerative Medicine, Guangzhou Institutes of Biomedicine and Health, Guangzhou, China. [4]University of Chinese Academy of Sciences, Beijing, China. [5]BGI Research, Shenzhen, China. [6]Joint School of Life Sciences, Guangzhou Institutes of Biomedicine and Health, Chinese Academy of Sciences, Guangzhou Medical University, Guangzhou, China. [7]Stem Cell Research Facility, Sloan Kettering Institute, New York, NY, USA. [8]BGI Research, Hangzhou, China. [9]College of Life Sciences, University of Chinese Academy of Sciences, Beijing, China. [10]Department of Systems Biology, School of Life Sciences, Southern University of Science and Technology, Shenzhen, China. [11]These authors contributed equally: Yueli Yang, Wenqi Jia. ✉e-mail: miguelesteban@genomics.cn; mazid@gibh.ac.cn; li_wenjuan@gibh.ac.cn

been identified as a key upstream transcription factor involved in TE formation. TE-specific genes, such as *Cdx2*, *Gata3*, and *Fgfr2* fail to upregulate in *Tead4* knockout mice resulting in pre-implantation embryonic lethality[5–7]. TEAD4 activity is tightly controlled by the Hippo pathway[8–10]. At the onset of mouse blastocyst formation, Hippo signaling is inactive in the outer embryonic layer of cells that will form the TE, and so its downstream target protein YAP (encoded by *Yap1*) is in an unphosphorylated form. Unphosphorylated YAP localizes in the nucleus and partners with TEAD4 to induce TE specifiers including *Cdx2* and *Gata3*[11]. In the ICM, the Hippo pathway is activated and leads to YAP phosphorylation, promoting cytoplasmic retention and proteasomal degradation. The YAP-related protein WWTR1 (also called as TAZ, encoded by *Wwtr1*) plays a similar functional role to YAP in the early mouse embryo[12]. Intriguingly, recent studies have revealed critical differences between mouse and human in the regulation of the Hippo pathway in early development[13]. For example, YAP is localized in the nucleus of both human TE and ICM cells, suggesting that additional or alternative regulatory mechanisms are involved in human TE formation. Moreover, YAP activity promotes human naïve pluripotent stem cell (PSC) generation and self-renewal[14].

To explore early extraembryonic lineage specification, in vitro models have been developed. The derivation from early placenta samples of human trophoblast stem cells (TSCs), which can be used to model the villous cytotrophoblast of the human first-trimester placenta, provides a valuable tool for studying placenta formation[15]. This TSC derivation medium can also be applied to induce TSCs from human naïve PSCs, which mimic pre-implantation ICM cells[16–23]. Primed human PSCs, which mimic post-implantation epiblast cells, can acquire cytotrophoblast-like features using either BMP4 alone or in combination with small molecules (*e.g.*, A83-01 and PD173074)[24,25]. However, the resulting cells only capture some cytotrophoblast characteristics and instead more closely resemble the amnion[26,27]. Primed human PSCs can also differentiate into functional TSCs with modified culture media[22,28,29], but it is unclear whether this involves the same path as naïve PSCs. Recent reports have used these models to gain insight into early human placenta specification. For example, primate-specific miRNA on chromosome 19 (C19MC) and the transcription factors TEAD1 and TEAD4 are essential for producing human TSCs[30–32]. Mechanisms related to the onset of TE specification and most aspects of human TSC induction and maintenance remain in general poorly understood.

In this study, we used human naïve PSCs generated with our recently developed 4CL culture medium[33] to produce TE-like cells (TELCs) and TSCs. We identified the transcriptional co-factor VGLL1 (vestigial-like family member 1) as a key regulator for the conversion of TELCs and maintenance of TSCs. VGLL1 has been previously reported to be highly expressed in the human placenta and is involved in preeclampsia[34,35]. Moreover, it can form a complex with TEAD4 in cancer cells[36–38]. We showed that in TELCs and TSCs, VGLL1 works through the interaction with TEAD4 by inducing histone H3 lysine 27 acetylation (H3K27ac) at target loci including cell cycle genes and TE/TSC-related genes. Besides, we found that VGLL1 and TEAD4 cooperate with GATA3 and TFAP2C. Importantly, VGLL1 is highly expressed in primate TE but negligibly in mouse, highlighting a potential interspecies difference in early placenta specification. We have thus uncovered a key regulatory mechanism of human early placenta formation whose further characterization may facilitate a deeper understanding of both normal and abnormal developmental processes.

## Results

### The transcriptional co-factor VGLL1 is highly expressed during primate TE induction

We produced TELCs from human naïve H9 embryonic stem cells (ESCs) generated with our newly established 4CL medium[33] using a previously reported TELC conversion protocol[26] (Fig. 1a). Flattened epithelial-like cells with a TE-like morphology were observed at day 5 (Fig. 1b). Downregulation of classical (*e.g.*, *NANOG* and *OCT4*) and naïve (*e.g.*, *KLF17* and *DPPA3*) pluripotency genes, expression of naïve/TE shared genes such as *TEAD4* and *TFAP2C*, and upregulation of TE-enriched genes like *GATA3* and *ENPEP* were validated by immunofluorescence microscopy and/or bulk RNA-sequencing (RNA-seq) (Fig. 1c, d and Supplementary Fig. 1a). To assess the fidelity with which TELCs derived from 4CL ESCs reflect preimplantation embryo TE cells, we performed droplet-based single-cell RNA sequencing (scRNA-seq)[39] at day 5 of TELC induction (TELC-D5) from 4CL ESCs and integrated them with previously reported scRNA-seq datasets of 4CL naïve ESCs[33], human preimplantation[40,41] and postimplantation embryo cells[42]. Uniform manifold approximation and projection (UMAP) representation showed that TELC-D5 cells clustered closely with human preimplantation embryo TE cells and their gene expression pattern was comparable (Fig. 1e, f). Pearson correlation analysis among the subclusters within TELC-D5 cells showed that they are highly correlated (correlation score > 0.9) except for two subclusters (cluster 7 and 8, 2.08 % of total TELC-D5 cells); cluster 7 also expressed SCT related genes (Supplementary Fig. 1b-d). These results suggest that 4CL ESC-derived TELC-D5 cells are relatively homogenous and resemble the human preimplantation embryo TE. We then determined the differentially expressed genes (DEGs) between naïve ESCs and TELC-D5 cells, and found *VGLL1* as the most significantly upregulated gene in TELCs (Fig. 1g). In contrast, *YAP1* and *TEAD4* were only moderately upregulated in TELC-D5 cells. We confirmed the exclusivity of *VGLL1* expression in TELCs by immunofluorescence microscopy, real-time quantitative PCR (RT-qPCR) and Western blotting (Fig. 1c, h, i). Western blotting also validated the downregulation of the pluripotency genes *NANOG*, *OCT4* and *SOX2* (Fig. 1i). Likewise, it confirmed that YAP and TEAD4 are slightly upregulated in TELC-D5 cells compared with naïve ESCs.

The high levels of *VGLL1* in TELCs derived from naive ESCs prompted us to study its expression kinetics during early in vivo development. The human early blastocyst corresponding to embryonic day 5 (E5) is comprised of cells belonging to the earliest stage of TE and ICM lineage. We investigated the identities and gene expression patterns of human E5 cells from a previously reported scRNA-seq dataset[40]. UMAP segregated the TE and ICM cells in addition to pre-lineage cells in E5 cells (Fig. 2a). Classical and naive pluripotency genes were highly expressed in pre-lineage and ICM cells but downregulated in TE cells (Supplementary Fig. 1e). TE-related transcription factors like *GATA3*, *GATA2* and *CDX2* were upregulated in TE cells compared with pre-lineage or ICM cells (Fig. 2b, Supplementary Fig. 1e). *VGLL1* was highly and specifically expressed in TE cells. This is relevant because context-dependent highly expressed genes often function as key regulators[43]. Unlike mouse, genes like *ELF5* and *EOMES* were only marginally expressed at this stage (Supplementary Fig. 1e), which is consistent with previous report[44]. To better understand the dynamic changes of gene expression during the human TE and ICM lineage bifurcation, we performed pseudotime analysis of the scRNA-seq data using Monocle 2[45–47] (Fig. 2c). As expected, pluripotency genes such as *OCT4* and *SOX2* increased in the pre-lineage to ICM branch and were downregulated in the pre-lineage to TE branch (Supplementary Fig. 1f). In contrast, naive/TE shared genes like *TEAD4*, *YAP1* and *TFAP2C* remained at similar expression levels along the pseudotime or were moderately upregulated, whereas TE-enriched genes like *GATA3*, *GATA2* and *CDX2* were progressively upregulated (Fig. 2d, Supplementary Fig. 1f). Further supporting a role in TE specification, *VGLL1* showed a dramatic upregulation at the onset of the TE branch (Fig. 2d). The quick induction and highly selective expression of *VGLL1* in TE were further confirmed by using another human early embryo scRNA-seq dataset[41] (Supplementary Fig. 1g-j).

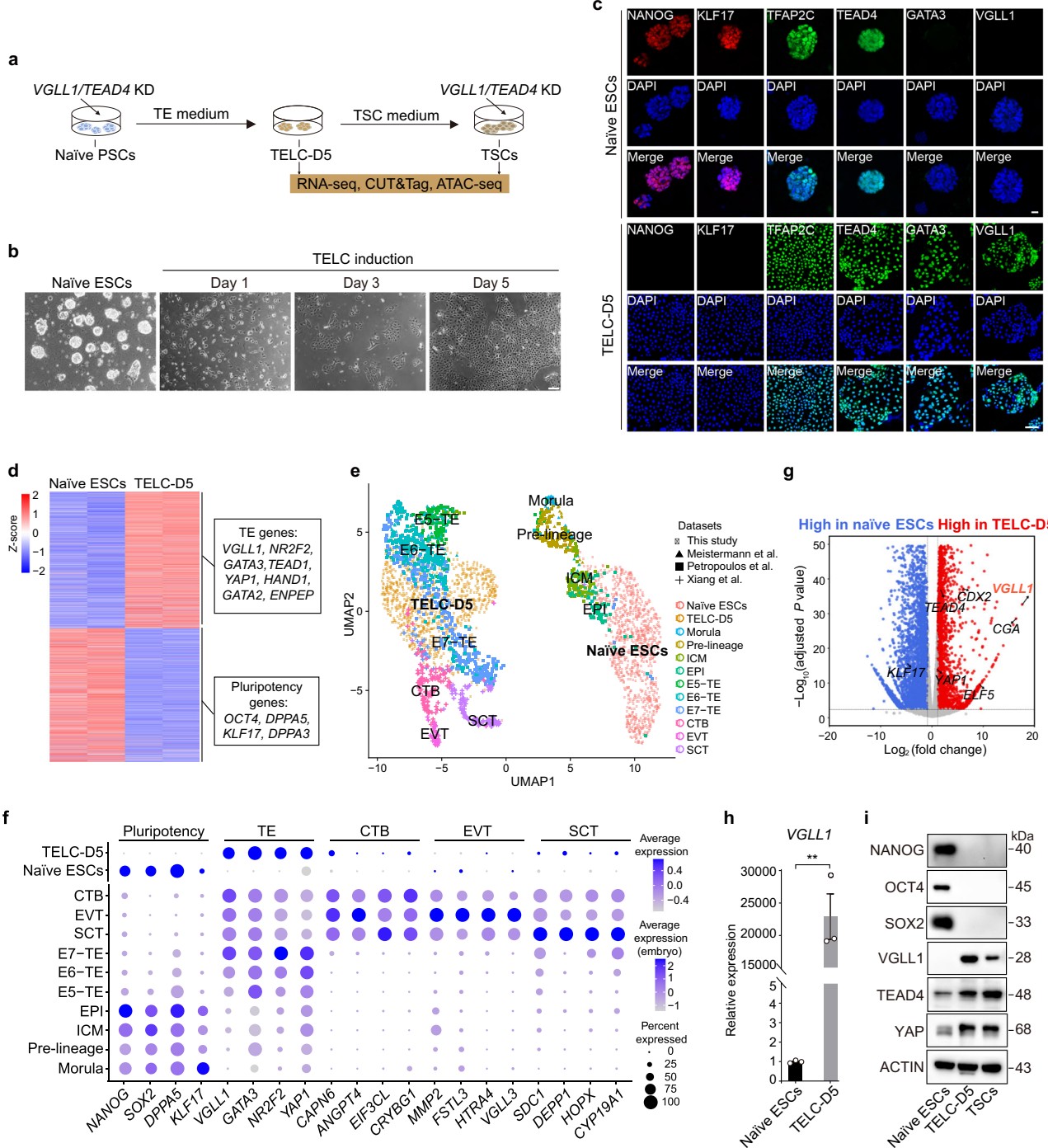

**Fig. 1 | VGLL1 is highly expressed during human TELC induction from naive PSCs. a** Schematic depicting the generation of TELC-D5 from 4CL H9 ESCs with TE medium and the generation of TSCs from TELC-D5 cells with TSC medium. *VGLL1/TEAD4* knockdown (KD) and multiple types of sequencing were performed as indicated. **b** Phase contrast images of 4CL H9 ESCs and TELCs at the indicated time points. Scale bar, 40 μm. Representative of three independent experiments. **c** Immunostaining images for pluripotency (NANOG and KLF17), naive/TE shared (TFAP2C and TEAD4), and TE-enriched (GATA3 and VGLL1) genes in 4CL H9 ESCs (upper panels; scale bar, 50 μm) and TELC-D5 cells (lower panels; scale bar, 100 μm). Nuclei were counterstained with DAPI (blue). Representative of three independent experiments. **d** Heatmap showing the expression of pluripotency and TE genes in bulk RNA-seq for the indicated conditions. Example genes for each cluster are shown in the boxes. *n* = 2 biological replicates. **e** UMAP comparing the human embryonic day 4 (E4) to E14 stages with 4CL H9 ESCs and TELC-D5 cells. All

reference datasets used in this study are summarized in Methods. **f** Bubble plot showing the frequency of expression and scaled average expression of representative genes in, 4CL H9 ESCs, TELC-D5 cells and human embryo E4-E14 datasets. TE, trophectoderm; CTB, cytotrophoblast; EVT, extravillous trophoblast; SCT, syncytiotrophoblast. **g** Volcano plot showing DEGs between 4CL H9 naive ESCs and TELC-D5 cells. DEGs higher in TELC-D5 cells (log₂(fold change) > 1) are shown in red. *P* value was calculated using the Wald test and adjusted for multiple testing using the Benjamini-Hochberg correction. **h** RT-qPCR showing the expression of *VGLL1* in 4CL H9 ESCs and TELC-D5 cells. Data are presented as the mean ± standard error of the mean (SEM) of the fold-change compared to naive ESCs. *n* = 3 biological replicates. *P* value was calculated using a two-tailed unpaired Student's *t*-test, ***P < 0.001. **i** Western blotting analysis for the indicated proteins in 4CL H9 ESCs, TELC-D5 cells, and TSCs derived from TELC-D5 cells. Representative of three independent experiments.

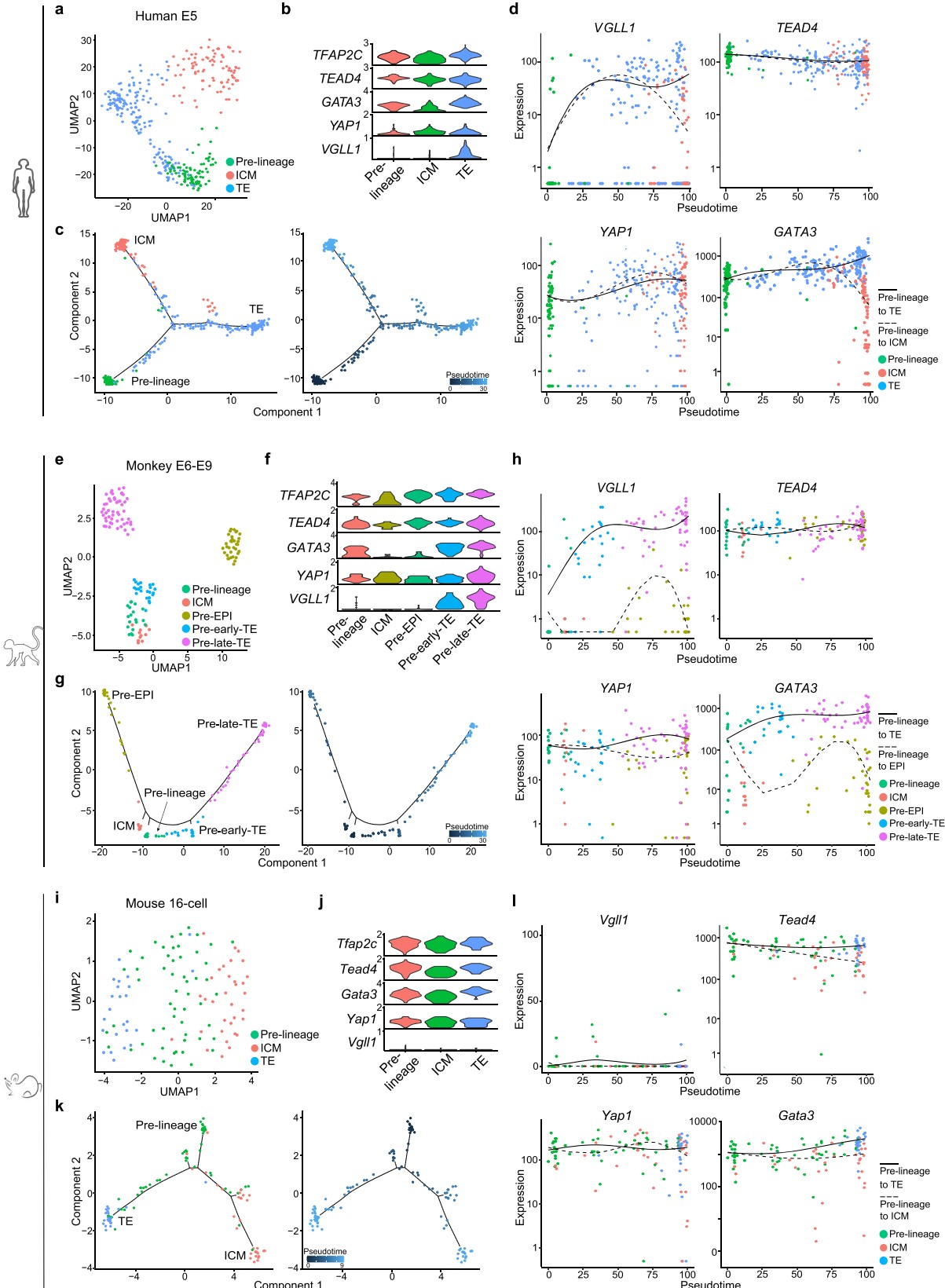

In addition, we investigated the cell types and gene expression pattern of early *Macaca fascicularis* monkey embryo using a reported scRNA-seq dataset[48] and found that, similar to human, *VGLL1* is highly expressed in a TE-specific manner (Fig. 2e, f). Pseudotime analysis also showed that *VGLL1* is strongly induced since the beginning of TE lineage specification (Fig. 2g, h). In contrast, scRNA-seq of early

mouse embryo development[49] showed that *Vgll1* is marginally expressed and minimally upregulated at the onset of mouse TE induction in the 16-cell stage (Fig. 2i-l), which is consistent with previous report that VGLL1 is not involved in mouse trophoblast specification[34]. These results indicate that VGLL1 may function as a TE induction regulator in primates.

**Fig. 2 | Expression kinetics of VGLL1 in mammalian preimplantation embryos.**
**a** UMAP visualization showing the different cell types in the human blastocyst at
E5[40]. Dots are colored by cell type. Cell types were annotated by cluster-specific
gene expression patterns. **b** Violin plot showing the log-normalized expression of
TE genes in different cell types of the human blastocyst at E5. **c** Left panel: the
trajectory of the different cell types in the human blastocyst at E5 reveals two
branches: pre-lineage to the TE branch and pre-lineage to the ICM branch. Right
panel: colors from dark blue to light blue indicate progression through the pseu-
dotime. **d** Expression patterns of TE genes along the pseudotime trajectory of cells
in the human blastocyst at E5. **e** UMAP visualization showing the different cell types
in the monkey blastocyst at E6-E9[48]. Dots are colored by cell type. Cell types were
annotated by cluster-specific gene expression patterns. **f** Violin plot showing the
log-normalized expression of TE genes in different cell types of the monkey
blastocyst at E6-E9. **g** Left panel: the trajectory of the different cell types in the
monkey blastocyst (E6-E9) reveals two branches: pre-lineage to the TE branch and
pre-lineage to the EPI branch. Right panel: colors from dark blue to light blue
indicate progression through the pseudotime. **h** Expression patterns of TE genes
along the pseudotime trajectory of cells in the monkey blastocyst at E6-E9. **i** UMAP
visualization showing the different cell types at the mouse 16-cell stage[49]. Dots are
colored by cell type. Cell types were annotated by cluster-specific gene expression
patterns. **j** Violin plot showing the log-normalized expression of TE genes in dif-
ferent cell types at the mouse 16-cell stage. **k** Left panel: the trajectory of the
different cell types at the mouse 16-cell stage reveals two branches: pre-lineage to
the TE branch and pre-lineage to the ICM branch. Right panel: colors from dark blue
to light blue indicate progression through the pseudotime. **l** Expression patterns of
TE genes along the pseudotime trajectory of cells at the mouse 16-cell stage.

## VGLL1 regulates TELC generation from human naive PSCs

To explore whether *VGLL1* has a functional role in human TE specifi-
cation, we depleted it using short-hairpin RNAs (shRNAs; sh*VGLL1*−2
and −4) during the 4CL naive PSC to TELC induction. Knockdown
efficiency was confirmed by RT-qPCR and Western blotting (Fig. 3a, b).
Notably, we observed that self-renewal (cell proliferation) during the
conversion was severely compromised in sh*VGLL1* transduced cells
compared to the sh*Luc* control, and this was also accompanied by a
lack of TE-like morphology (Fig. 3c, d). We then performed bulk RNA-
seq for sh*Luc* and sh*VGLL1* transduced TELC-D5 cells. Consistent with
the reduced proliferation and lack of TE-like morphology, cell-cycle/
self-renewal genes such as *MKI67*, *CDK1/3/7* and *CDC42*, and TE genes
like *GATA3, ENPEP, NR2F2* and *TEAD1,* were substantially down-
regulated in sh*VGLL1* transduced cells compared to control sh*Luc*
(Fig. 3e). There was relatively less downregulation of pluripotency
genes in sh*VGLL1* cells compared to sh*Luc* control (Fig. 3e). These
results were validated by RT-qPCR (Fig. 3f). In agreement, gene
ontology (GO) analysis for the downregulated genes with sh*VGLL1*
showed enrichment of terms related to cell cycle DNA replication, cell
cycle checkpoint, *in utero* embryonic development and placenta
development (Fig. 3g). Likewise, upregulated genes in sh*VGLL1* trans-
duced cells compared to control sh*Luc* were related to development
and cell fate commitment (Fig. 3h). To further validate the function of
*VGLL1* in TELC induction, we generated *VGLL1* knockout (KO) clones
(C45 and C68) using CRISPR-Cas9 technology with H9 ESCs (Fig. 3i, j).
Consistent with the shRNA-mediated *VGLL1* knockdown, both *VGLL1*
KO clones, compared to wild-type (WT) clones, showed reduced TE
gene expression determined by RNA-seq as well as compromised cell
proliferation confirmed by phase contrast imaging and cell counting
(Fig. 3k–m). The downregulated genes in *VGLL1* KO cells were asso-
ciated with *in utero* embryonic development, placenta development
and positive regulation of cell cycle determined by GO enrichment
analysis (Fig. 3n).

Trophoblast lineage differentiation potential may vary with cell
lines and starting naive PSC states[17]. We thus performed *VGLL1*
knockdown in another human induced pluripotent stem cell (iPSC)
line, UH10[50,51], and tested other well-established human naive PSC
culture conditions: PXGL and HENSM[52,53]. Consistently, *VGLL1* knock-
down during TELC induction resulted in reduced TE gene expression
and compromised cell proliferation in UH10 iPSCs in 4CL medium or
H9 ESCs in PXGL and HENSM media (Supplementary Fig. 2a-i). These
findings demonstrate that *VGLL1* is required for the induction of
human TELCs regardless of the cell line or starting naive PSC medium.

To exclude the possibility that the effect of *VGLL1* depletion on TE
gene expression is due to cell cycle arrest, we generated two effective
shRNA lentiviruses targeting *TP53* (sh*TP53*−1 and sh*TP53*−2) and then
transduced them into WT and *VGLL1* KO (C45 and C68) H9 ESCs, along
with sh*Luc* as control (Supplementary Fig. 2j). In all settings, we
observed a 50-70% increase of cell number in *TP53* knockdown cells
relative to sh*Luc* control cells, as expected (Supplementary Fig. 2k).
Despite the increased cell number, we did not observe upregulation of
TE genes like *GATA2, GATA3, TFAP2C* and *YAP1* in *TP53* knockdown cells
compared with sh*Luc* control cells, providing evidence that the reg-
ulatory function of VGLL1 on TE gene expression is independent of cell
cycle arrest (Supplementary Fig. 2l).

Therefore, we have demonstrated that *VGLL1* is a critical regulator
of the human naive PSC to TELC transition, highlighting a major dif-
ference with mouse.

## VGLL1 works in tandem with TEAD4 to regulate human TELC specification

To dissect the function of VGLL1, we performed Cleavage Under Tar-
gets and Tagmentation (CUT&Tag)[54] for VGLL1 in TELC-D5 cells dif-
ferentiated from 4CL H9 ESCs to determine its genomic occupancy. A
total of 54,773 binding loci were captured (Supplementary Data 1).
Peak distribution analysis revealed that VGLL1 primarily binds to pro-
moter, distal intergenic and intronic regions (Fig. 4a). This occupancy
pattern suggested that VGLL1 regulates target genes by controlling
both promoter and enhancer activity. Consistently, cross-comparison
with our bulk RNA-seq results showed that 78.2% of downregulated
genes in sh*VGLL1* compared with sh*Luc* are bound by VGLL1 (Fig. 4b),
supporting a direct regulatory effect of VGLL1 on these genes. Genes
bound by VGLL1 were enriched for GO terms such as placenta devel-
opment, metaphase/anaphase transition of mitotic cell cycle, and
cilium assembly (Fig. 4c), among others. This shows that VGLL1 reg-
ulates genes involved in multiple aspects of TE function.

VGLL1 is a transcriptional co-factor[55] and is not predicted to bind
DNA directly. Therefore, we performed motif discovery analysis to
uncover its potential transcription factor partners. The DNA-binding
motif for TEAD4, a well-known transcription factor necessary for TE
specification in mouse[5,7], was highly enriched in VGLL1-bound sites
(Fig. 4d). We then asked whether VGLL1 physically interacts with
TEAD4. We first did a reciprocal co-immunoprecipitation in two cell
lines (H9 ESCs and UH10 iPSCs) at day 5 of TELC induction from 4CL
PSCs and found that VGLL1 and TEAD4 strongly interact with each
other (Fig. 4e and Supplementary Fig. 3a). Moreover, in line with pre-
vious reports[32], there was reduced cell proliferation and compromised
TELC identity when we knocked down or knocked out *TEAD4* during
the TELC conversion, irrespective of the cell lines or naive culture
conditions (Supplementary Fig. 3b–r). We also confirmed that the
effect of TEAD4 suppression on TE gene expression is not due to cell
cycle arrest using 4CL H9 ESCs (Supplementary Fig. 4a–c). These
results demonstrate that VGLL1 forms a complex with TEAD4 to reg-
ulate human TELC induction.

TEAD4 controls mouse and human trophoblast lineage specifi-
cation by interacting with nuclear YAP and WWTR1 to regulate target
gene expression[11,56]. To explore the potential functional relationship of
different co-factors with TEAD4, we examined the interaction of
TEAD4 with YAP and WWTR1 through co-immunoprecipitation in 4CL
H9 ESCs and UH10 iPSCs. We observed that TEAD4 also interacts with
them, but to a much lesser extent than with VGLL1 (Fig. 4e, Supple-
mentary Fig. 3a). Notably, VGLL1 did not interact with YAP or WWTR1.

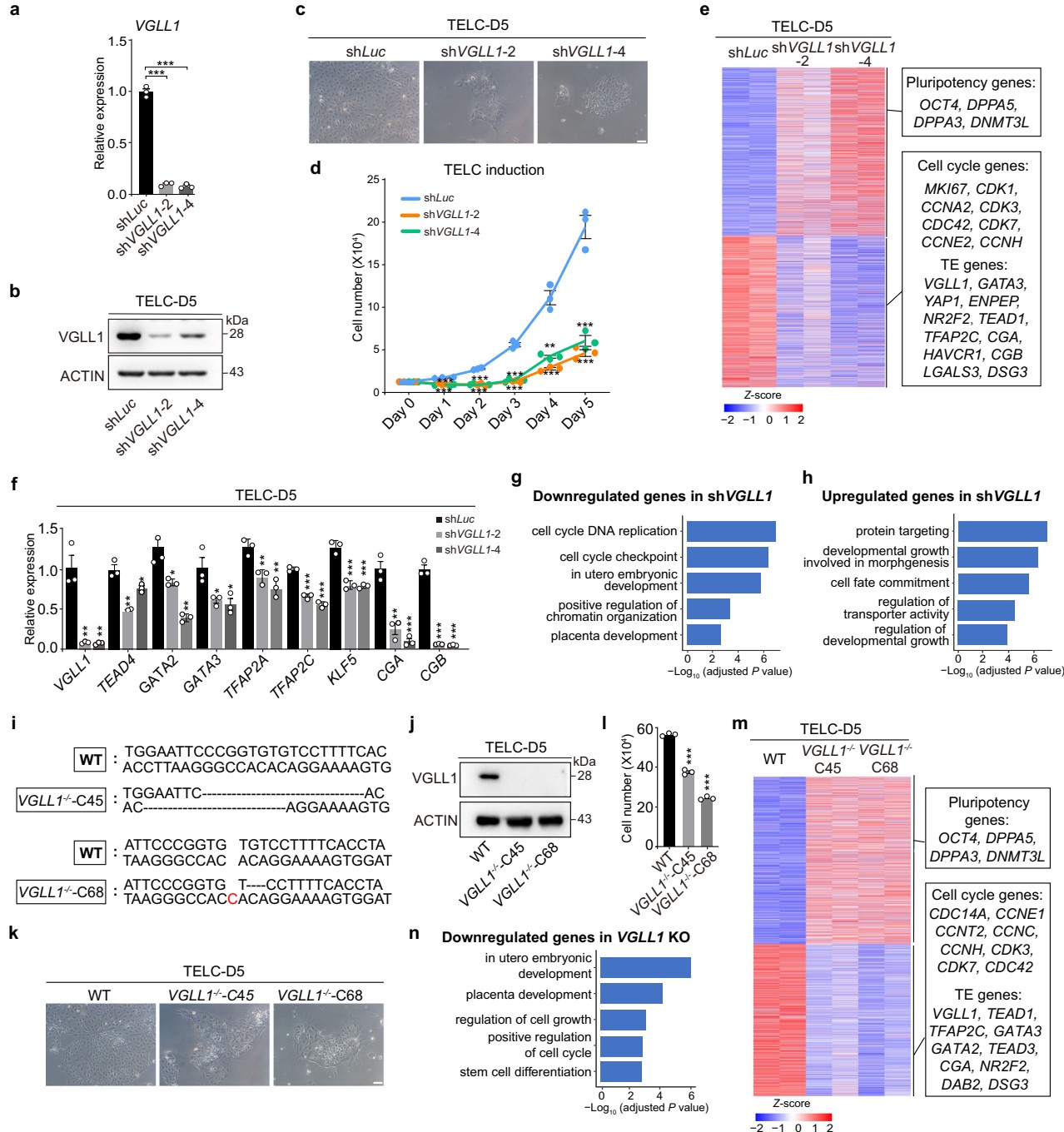

Consistently, the knockdown of both *YAP1 or WWTR1* hindered TELC induction (Supplementary Fig. 4d–k), indicating that TEAD4 recruits all three co-factors to regulate human TELC induction.

To further illuminate how VGLL1 works in tandem with TEAD4 along with other co-factors like YAP, we performed CUT&Tag for TEAD4 and YAP in TELC-D5 cells differentiated from 4CL H9 ESCs. Genome-wide binding analysis revealed that regions bound by TEAD4 and YAP were also largely located at the promoter, distal intergenic and intronic regions (Fig. 4f, g). Co-occupancy analysis between VGLL1, YAP and TEAD4 showed that VGLL1 and TEAD4 share a large number (43,048) of binding sites, among which 57 % were also co-bound by YAP (24,626) (Fig. 4h). This data correlates with the co-immunoprecipitation results indicating that TEAD4 interacts more strongly with VGLL1 than with YAP (Fig. 4e, Supplementary Fig. 3a). Genes co-bound by VGLL1, TEAD4 and YAP included cell cycle/self-

renewal genes and TE-related genes (Fig. 4h). We then performed GO analysis for genes co-bound by all these three factors, and those bound by either two of them in TELC-D5 cells. VGLL1-TEAD4 only or VGLL1-TEAD4-YAP co-bound genes were enriched in terms related to placenta development, cell cycle regulation and, interestingly, histone acetylation, while YAP-TEAD4 only co-bound genes were enriched for other biological processes such as protein poly-ubiquitination and positive regulation of autophagy (Fig. 4i). The distribution of these three factors across the genome indicates that VGLL1 and YAP have both shared and unique functions. We also tested the subcellular localization of VGLL1, TEAD4 and YAP in TELC-D5 cells derived from 4CL H9 ESCs. Subcellular fractionation followed by Western blotting showed that VGLL1 and TEAD4 mainly localized in the nucleus (Supplementary Fig. 4l), while YAP was more restricted to the cytoplasm. The distinct subcellular distribution

**Fig. 3 | VGLL1 is required for human TELC induction from naive PSCs. a** RT-qPCR showing the *VGLL1* knockdown efficiency for 4CL H9 ESCs transduced with sh*VGLL1*−2 or sh*VGLL1*−4 compared to the sh*Luc* control at day 5 of TELC differentiation. Data are presented as the mean ± SEM, n = 3 biological replicates. *P* value was calculated using a two-tailed unpaired Student's *t*-test, ***P* < 0.001. **b** Western blotting analysis of the indicated proteins for 4CL H9 ESCs transduced with sh*VGLL1*−2 or sh*VGLL1*−4 compared to the sh*Luc* control at day 5 of TELC differentiation. Representative of three independent experiments. **c** Representative phase contrast images of 4CL H9 ESCs transduced with sh*Luc* (control) or sh*VGLL1* (2 and 4) at day 5 of TELC differentiation. Scale bar, 100 μm. Representative of three independent experiments. **d** Analysis of cell numbers for 4CL H9 ESCs transduced with sh*Luc*, sh*VGLL1*−2 or sh*VGLL1*−4 throughout the TELC induction time course. Data are presented as the mean ± SEM. n = 3 biological replicates. *P* value was calculated using a two-tailed unpaired Student's *t*-test, ***P* < 0.001, **P* < 0.01. **e** Heatmap showing the expression of pluripotency, cell cycle and TE genes in 4CL H9 ESC-derived cells from the indicated conditions. Example genes are shown for each cluster in the boxes. n = 2 biological replicates. **f** RT-qPCR showing the expression of TE-related genes for 4CL H9 ESCs transduced with sh*VGLL1*−2 and sh*VGLL1*−4 compared to the sh*Luc* control at day 5 of TELC differentiation. Data are presented as the mean ± SEM. n = 3 biological replicates. *P* value was calculated using a two-tailed unpaired Student's *t*-test, ***P* < 0.001, **P* < 0.01, *P* < 0.05. **g** Enriched GO terms for downregulated genes in 4CL H9 ESCs transduced with

sh*VGLL1* compared to sh*Luc* control at day 5 of TELC differentiation. *P* value was calculated using a hypergeometric test (one-sided) and adjusted for multiple testing using the Benjamini-Hochberg correction. **h** Enriched GO terms for upregulated genes in 4CL H9 ESCs transduced with sh*VGLL1* compared to sh*Luc* control at day 5 of TELC differentiation. *P* value was calculated using a hypergeometric test (one-sided) and adjusted for multiple testing using the Benjamini-Hochberg correction. **i** TA cloning followed by Sanger sequencing results showing homozygous deletion for *VGLL1*-knockout clones [clone 45 (C45) and clone 68 (C68)]. WT: wild-type. **j** Western blotting analysis of VGLL1 expression in H9 WT and *VGLL1*-knockout clones at day 5 of TELC induction. Representative of three independent experiments. **k** Representative phase contrast images of H9 WT and *VGLL1*-knockout clones at day 5 of TELC induction. Scale bar, 100 μm. Representative of three independent experiments. **l** Analysis of cell numbers for H9 WT and *VGLL1*-knockout clones at day 5 of TELC induction. Data are presented as the mean ± SEM. n = 3 biological replicates. *P* value was calculated using a two-tailed unpaired Student's *t*-test, ***P* < 0.001. **m** Heatmap showing the expression of pluripotency, cell cycle and TE genes in bulk RNA-seq of H9 *VGLL1*-knockout clones compared to WT at day 5 of TELC induction. Example genes are shown for each cluster in the boxes. n = 2 biological replicates. **n** Enriched GO terms for downregulated genes in H9 *VGLL1*-knockout clones compared to WT at day 5 of TELC induction. *P* value was calculated using a hypergeometric test (one-sided) and adjusted for multiple testing using the Benjamini-Hochberg correction.

---

further supports that TEAD4 interact less with YAP compared to VGLL1.

These results demonstrate that VGLL1 functions as a strong co-factor for TEAD4 in addition to YAP and WWTR1, potentially adding more flexibility to safeguard TE development in human.

## VGLL1 controls human TELC induction by promoting chromatin remodeling at target loci

We next aimed to elucidate the underlying mechanism by which VGLL1 and TEAD4 regulate the expression of TE genes. Gene expression is highly dependent on the enrichment of active histone marks at target loci and in recent years histone acetylation has emerged as a major regulator of cell (including PSC) fate transitions[57]. We thus tested whether *VGLL1* knockdown affects histone acetylation in TELCs derived from 4CL H9 ESCs and UH10 iPSCs. Western blotting showed that total histone H3 acetylation (H3ac) and histone H4 acetylation (H4ac) levels were decreased upon *VGLL1* depletion in TELC-D5 cells compared to the sh*Luc* control (Fig. 5a and Supplementary Fig. 5a). Importantly, histone H3 lysine 27 acetylation (H3K27ac), which often marks promoters and enhancers of active genes, was also reduced upon *VGLL1* knockdown. This finding suggested that VGLL1 is involved in H3K27ac deposition. Inspection of the sites co-occupied by VGLL1 and TEAD4 and H3K27ac showed a high degree of overlap, with 73.3% percent of VGLL1-TEAD4 co-bound sites displaying enriched H3K27ac (Fig. 5b). These loci included genes related to self-renewal and TE specification/function genes. Enriched GO terms corresponding to these shared loci belonged to categories related to placenta development and cell cycle (Fig. 5c).

To better understand the relationship between TEAD4 and VGLL1 co-binding with H3K27ac enrichment, we also performed CUT&Tag for TEAD4 and H3K27ac in naïve PSCs. We first compared the TEAD4 binding pattern between 4CL H9 ESCs and TELC-D5 cells and identified three categories (naïve-specific-, shared- and TE-specific loci) of TEAD4-binding peaks (Fig. 5d). Naïve ESC-specific TEAD4-bound genes included pluripotency transcription factors such as *OCT4*, *SOX2*, *KLF17* and *DPPA5*, consistent with the observation that TEAD4 buffers pluripotency genes in human PSCs[58]. Shared TEAD4-bound genes included many cell-cycle regulators and naïve/TE-shared genes such as *CDK1*, *MKI67*, *TEAD4* and *TFAP2C*. As expected, the largest number of genes corresponded to TE-specific TEAD4-bound genes, which included many TE-enriched genes like *VGLL1*, *GATA3*, *CDX2* and *NR2F2*. Next, we performed a signal density pile-up analysis for H3K27ac (naïve ESCs

and TELC-D5 cells) and VGLL1 (TELC-D5 cells) occupancy based on the categories of the identified TEAD4-binding peaks (Fig. 5e). Notably, we observed that H3K27ac levels in naïve-specific TEAD4-bound genes were high in naïve ESCs but low in TELC-D5 cells. In contrast, H3K27ac levels in shared and TE-specific TEAD4-bound genes were increased in TELC-D5 cells, being more obviously in the latter case.

Because high H3K27ac levels influence target gene expression at least in part by increasing local chromatin accessibility[59], we also did an assay for transposase-accessible chromatin using sequencing (ATAC-seq) for TELC-D5 cells differentiated from 4CL H9 ESCs. This showed that 95.27% of the loci co-bound by VGLL1-TEAD4 and enriched with H3K27ac display accessible chromatin (Fig. 5f, g). The regions with overlapping peaks included genes involved in placenta development and cell cycle regulation (Fig. 5h, i and Supplementary Fig. 5b, c). On the contrary, open chromatin regions only bound by VGLL1 or TEAD4 were mostly related with other lineages (Supplementary Fig. 5d-f). Of note, VGLL1 and TEAD4 were also bound by themselves and enriched for H3K27ac, indicating a positive feedback regulation (Fig. 5i and Supplementary Fig. 5c). To provide more evidence of the functional cooperation between VGLL1 and TEAD4 and the role of VGLL1 in histone acetylation, we measured TEAD4 binding and H3K27ac enrichment using CUT&Tag and chromatin accessibility using ATAC-seq in *VGLL1* KO cells at day 5 of TELC induction from 4CL H9 ESCs. In *VGLL1* KO cells, the average genome binding intensity of TEAD4 and the enrichment of H3K27ac at VGLL1-binding sites declined compared to WT cells (Fig. 5j, k). Along with the decrease of H3K27ac marks, the chromatin accessibility at these regions was also reduced in *VGLL1* KO cells (Fig. 5l).

Overall, we have demonstrated that TEAD4 becomes redistributed in TELCs compared to naïve PSCs and highlighted a link between VGLL1 recruitment by TEAD4 at target loci and chromatin opening through increased histone acetylation.

## VGLL1 is necessary to sustain the expandable human TSC identity

Upon implantation, the TE develops into the mononucleated cytotrophoblast. Cytotrophoblast cells self-renew while simultaneously produce the multinucleated syncytiotrophoblast (SCT) cells and the highly invasive extravillous trophoblast (EVT) cells, both of which are critical for forming the placenta. To further enhance our understanding about the role of VGLL1 in early human placenta development, we tested whether it also functions in maintaining TSC identity.

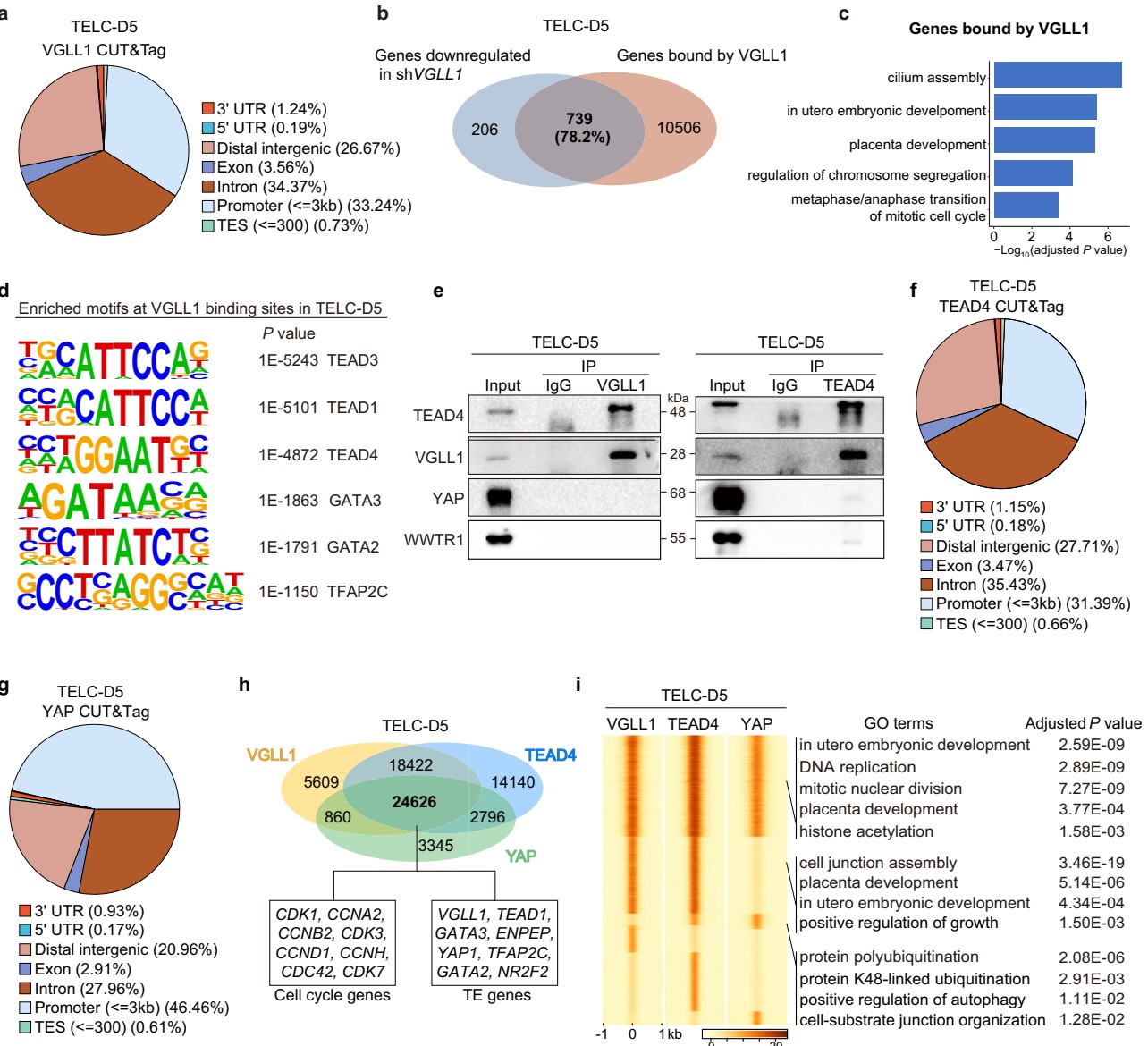

**Fig. 4 | VGLL1 interacts with TEAD4 to regulate human TELC induction from naïve PSCs. a** Pie chart showing the distribution of VGLL1-binding sites throughout the annotated genomic locations in CUT&Tag of TELC-D5 cells differentiated from 4CL H9 ESCs. **b** Venn diagram showing the overlap between downregulated genes in sh*VGLL1* compared to sh*Luc* control and genes bound by VGLL1 at promoter sites in TELC-D5 differentiated from 4CL H9 ESCs. **c** Enriched GO terms for genes bound by VGLL1 in TELC-D5 cells differentiated from 4CL H9 ESCs. *P* value was calculated using a hypergeometric test (one-sided) and adjusted for multiple testing using the Benjamini-Hochberg correction. **d** Motif analysis using HOMER showing the significantly enriched DNA-binding motifs at VGLL1-bound sites in TELC-D5 cells differentiated from 4CL H9 ESCs. *P* value was calculated using a hypergeometric test (one-sided). **e** Immunoprecipitation using lysates from TELC-D5 cells differentiated from 4CL H9 ESCs with anti-VGLL1 (left panel) or anti-TEAD4 (right panel) and

subsequent Western blotting analysis with anti-TEAD4, anti-VGLL1, anti-YAP and anti-WWTR1. Representative of three independent experiments. **f** Pie chart showing the distribution of TEAD4-binding sites throughout the annotated genomic locations in TELC-D5 cells differentiated from 4CL H9 ESCs. **g** Pie chart showing the distribution of YAP-binding sites throughout the annotated genomic locations in TELC-D5 cells differentiated from 4CL H9 ESCs. **h** Venn diagram showing the overlap between VGLL1-, TEAD4- and YAP-bound sites in CUT&Tag of TELC-D5 cells differentiated from 4CL H9 ESCs. Representative cell cycle and TE genes corresponding to the overlapping regions are shown. **i** Co-occupancy analysis by signal density pileups of VGLL1, TEAD4 and YAP peaks in CUT&Tag of TELC-D5 differentiated from 4CL H9 ESCs. GO analysis for each group is shown. *P* value was calculated using a hypergeometric test (one-sided) and adjusted for multiple testing using the Benjamini-Hochberg correction.

We performed TSC derivation from 4CL H9 ESCs using a previously described medium[15]. TELC-D5 cells were cultured in this medium for another 5-7 days to convert them into stable and expandable TSCs, which expressed VGLL1, TEAD4, YAP and showed loss of pluripotency genes like NANOG, OCT4 and SOX2 at the protein level (Fig. 1a, Fig. 1i and Supplementary Fig. 6a). Consistent with the findings in TELCs, *VGLL1* was among the highest expressed genes in TSCs compared to naïve ESCs, as assessed by bulk RNA-seq and validated with RT-qPCR (Supplementary Fig. 6b-d). Likewise, *VGLL1* knockdown compromised

TSC self-renewal as detected by a change in cell morphology and reduced proliferation compared to the sh*Luc* control (Fig. 6a, b and Supplementary Fig. 6e, f). Bulk RNA-seq analysis also showed that cell cycle- and TSC-related (*e.g., GATA3, CAPN6, GCM1*) genes were downregulated in *VGLL1* depleted cells (Fig. 6c). GO analysis for downregulated genes in *VGLL1* knockdown cells included biological processes such as cell cycle regulation and placenta development (Fig. 6d). The need for *VGLL1* was further validated by performing shRNA knockdown in UH10 iPSCs and embryo blastocyst-derived TSCs

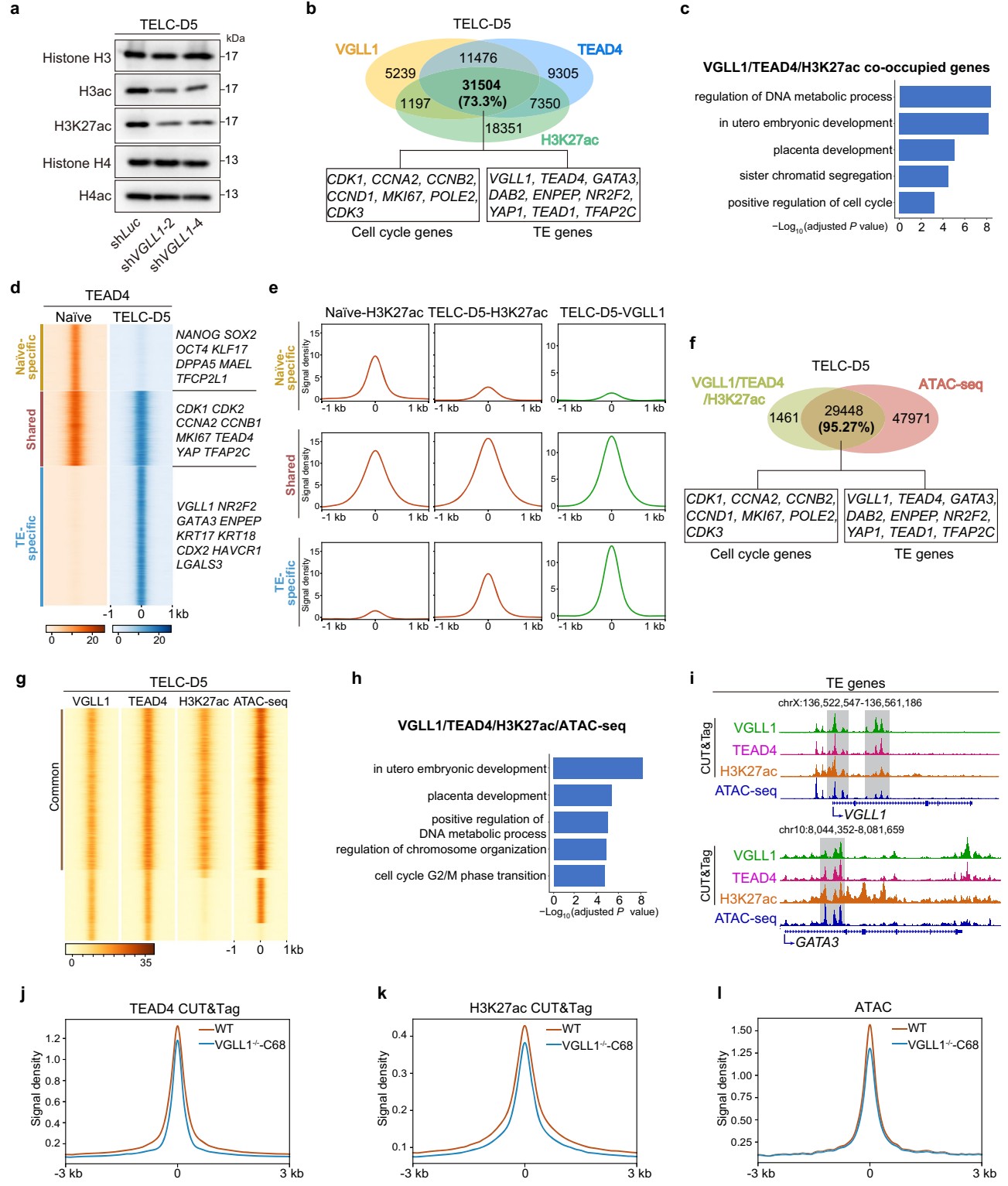

**a** TELC-D5

**b** TELC-D5

**c** VGLL1/TEAD4/H3K27ac co-occupied genes

**d** TEAD4

**e** Naïve-H3K27ac  TELC-D5-H3K27ac  TELC-D5-VGLL1

**f** TELC-D5

**g** TELC-D5

**h** VGLL1/TEAD4/H3K27ac/ATAC-seq

**i** TE genes

**j** TEAD4 CUT&Tag   **k** H3K27ac CUT&Tag   **l** ATAC

(TSC^BT)[15] (Supplementary Fig. 6g-l). To assess the functional role of *VGLL1* during SCT and EVT differentiation, expandable TSCs derived from TELCs differentiated from 4CL H9 ESCs were transduced with two effective shRNAs for *VGLL1* along with sh*Luc* as control. Subsequently, SCT and EVT differentiation were carried out at the time of shRNA transduction as previously described[60]. *VGLL1* knockdown resulted in reduced cell proliferation, widespread cell death and failure to differentiate into SCT or EVT cells, while the control (sh*Luc*) cells successfully differentiated and they expressed specific markers, SDC1 for SCT and HLA-G for EVT (Supplementary Fig. 6m, n). This

demonstrates the importance of *VGLL1* in maintaining the functional potential of TSCs.

To study the regulatory mechanism of VGLL1 in maintaining TSC identity, we first confirmed the interaction between VGLL1 and TEAD4 in TSCs derived from H9 ESCs and UH10 iPSCs by performing reciprocal co-immunoprecipitation (Fig. 6e and Supplementary Fig. 7a). Consistent with TELC-D5 cells, TEAD4 showed stronger interaction with VGLL1 compared with YAP and WWTR1, even though their knockdown also compromised TSC identity (Supplementary Fig. 7b-i). In addition, we studied whether the subcellular localization of VGLL1

**Fig. 5 | VGLL1 regulates histone acetylation levels and chromatin accessibility at TE target loci. a** Western blotting analysis for the indicated histone marks in 4CL H9 ESCs transduced with sh*Luc*, sh*VGLL1*−2 or sh*VGLL1*−4 at day 5 of TELC differentiation. Representative of three independent experiments. **b** Venn diagram showing the overlap between VGLL1- and TEAD4-bound sites and H3K27ac-enriched sites in CUT&Tag of TELC-D5 cells differentiated from 4CL H9 ESCs. Representative cell cycle and TE genes corresponding to the overlapping regions are shown in the boxes. **c** Enriched GO terms for VGLL1, TEAD4 and H3K27ac co-occupied genes in CUT&Tag of TELC-D5 differentiated from 4CL H9 ESCs. *P* value was calculated using a hypergeometric test (one-sided) and adjusted for multiple testing using the Benjamini-Hochberg correction. **d** Co-occupancy analysis by signal density pileups of TEAD4 CUT&Tag peaks in 4CL H9 ESCs and TEAD4 CUT&Tag peaks in TELC-D5 cells differentiated from them. Peaks were divided into three categories: naïve-specific, shared and TE-specific. Example genes are shown for each category. **e** Signal intensity of H3K27ac in 4CL H9 ESCs (Naïve-H3K27ac) and TELC-D5 cells (TELC-D5-H3K27ac), and VGLL1 in TELC-D5 cells (TELC-D5-VGLL1) from the three categories defined in **d**. **f** Venn diagram showing the overlap between VGLL1- and TEAD4-binding at H3K27ac-enriched sites in CUT&Tag with

chromatin accessibility peaks (ATAC-seq) in TELC-D5 cells differentiated from H9 4CL naïve ESCs. Representative cell cycle and TE genes corresponding to the overlapping regions are shown in the boxes. **g** Co-occupancy analysis by signal density pileups of VGLL1, TEAD4, H3K27ac CUT&Tag peaks and chromatin accessibility (ATAC-seq) peaks in TELC-D5 cells differentiated from 4CL H9 ESCs. **h** Enriched GO terms for open chromatin genes (ATAC-seq) co-occupied by VGLL1, TEAD4 and H3K27ac (CUT&Tag) of TELC-D5 cells differentiated from 4CL H9 ESCs. *P* value was calculated using a hypergeometric test (one-sided) and adjusted for multiple testing using the Benjamini-Hochberg correction. **i** Genome browser tracks showing VGLL1, TEAD4, H3K27ac genomic enrichment peaks (CUT&Tag) and chromatin accessibility peaks (ATAC-seq) for representative TE gene loci of TELC-D5 cells differentiated from 4CL H9 ESCs. **j** Signal intensity of TEAD4 at VGLL1-bound sites in WT and *VGLL1*-KO clone 68 (*VGLL1*[−/−]-C68) at day 5 of TELC induction from 4CL H9 ESCs. **k** Signal intensity of H3K27ac at VGLL1-binding sites in WT and *VGLL1*-KO clone 68 (*VGLL1*[−/−]-C68) at day 5 of TELC induction from 4CL H9 ESCs. **l** Signal intensity of chromatin openness at VGLL1-binding sites in WT and *VGLL1*-KO clone 68 (*VGLL1*[−/−]-C68) at day 5 of TELC induction from 4CL H9 ESCs.

and TEAD4 is similar to TELC-D5 cells. Unlike TELCs, VGLL1 was evenly distributed between the nucleus and cytoplasm in TSCs, while TEAD4 tended to localize more in the nucleus (Supplementary Fig. 7j). Additionally, we demonstrated that TEAD4 is important for preserving TSC identity by performing shRNA knockdown in TSCs derived from different PSC lines and TSC[BT] (Supplementary Fig. 7k–u). These results indicate that TEAD4 is also required to safeguard TSC differentiation potential (Supplementary Fig. 7v, w).

We then performed CUT&Tag for VGLL1 and TEAD4 in H9 TSCs and noticed that as in TELC-D5 cells, they share a large proportion (~80%) of genomic binding sites (Supplementary Fig. 8a). Similarly, TEAD4 was one of the most enriched motifs at VGLL1-bound sites (Supplementary Fig. 8b). We also noticed a decrease in the global levels of H3ac, H3K27ac and H4ac levels upon *VGLL1* knockdown in H9 ESCs and UH10 iPSCs (Fig. 6f and Supplementary Fig. 8c). A large proportion of VGLL1- and TEAD4-shared regions corresponded to H3K27ac CUT&Tag peaks (80.7%) and chromatin accessible regions detected by ATAC-seq (93.17% of VGLL1, TEAD4 and H3K27ac shared sites), and these genes were related to placenta development and cell cycle (Fig. 6g and Supplementary Fig. 8d–i).

To explore a potential functional difference of the VGLL1-TEAD4 complex in TELCs and TSCs, we compared the binding pattern of VGLL1 and TEAD4, which showed different groups. Group 1 peaks, which take up a substantial proportion of VGLL1 and TEAD4 co-bound loci, were shared between TELCs and TSCs (Fig. 6h). Group 2 peaks were co-bound by VGLL1 and TEAD4 only in TELCs, while group 3 were co-bound by VGLL1 and TEAD4 only in TSCs. Group 1 peaks included genes with functions mostly related to placenta development, cell cycle regulation, histone modification and especially histone acetylation (Fig. 6i), consistent with our functional study in TELCs and TSCs. Group 2 peaks corresponded to genes involved not only in placenta development but also in various other developmental processes such as mesonephric epithelium development and sensory system development. We speculate that binding of VGLL1 and TEAD4 at these genes correlates with an unstable preliminary extraembryonic cell fate, in which the chromatin regions of different lineage genes are still accessible for binding. In this regard, TE cells isolated from human blastocysts have the capacity to develop into ICM cells and express NANOG, suggesting that their identity is not yet fully committed[61]. Group 3 peaks comprised genes participating in signaling pathways and functional terms like phagocytosis and endocytosis. Consistent with the notion that VGLL1-TEAD4 co-bound sites are enriched with H3K27ac, we found that this active histone mark was also positively correlated with VGLL1-TEAD4 binding sites for group 2 and group 3. For example, genes like *APOB* and *GREM2* are bound by the VGLL1-TEAD4 complex only in TELCs (group 2) and their promoter and enhancer regions were also

enriched for H3K27ac (Fig. 6j and Supplementary Fig. 8j). On the contrary, genes like *PSG8* and *PSG1* were co-bound by VGLL1 and TEAD4 in TSCs (group 3), with higher adjacent H3K27ac levels.

We have demonstrated that the function of the tandem VGLL1-TEAD4 in controlling self-renewal and cell identity is conserved in both TELCs and TSCs derived from human naïve PSCs, although there are differences between both cell types.

## VGLL1-TEAD4 cooperate with GATA3 and TFAP2C to control the human TELC gene regulatory network

Cellular states are maintained by master gene regulatory networks (GRNs) consisting of key transcription factors and co-regulators[62]. It is known that the transcription factors GATA3 and TFAP2C are essential regulators of mouse and human TE specification[63,64]. TEAD4, GATA3 and TFAP2C are required for activating CDX2 expression and inducing TE lineage specification in mouse preimplantation embryos and GATA3 binds to the VGLL1 promoter to promote its expression in human trophoblast progenitors[65]. The regulatory network of TEAD4, GATA3 and TFAP2C in human models has not been studied in detail yet. To elucidate whether the VGLL1-TEAD4 complex acts synergistically with these two transcription factors, we performed CUT&Tag for GATA3 and TFAP2C in human TELC-D5 cells differentiated from 4CL H9 ESCs and compared it with the genome-wide binding of VGLL1 and TEAD4. Pairwise comparison showed that VGLL1 and TEAD4 shared 79.47% and 69.62%, respectively, of their binding sites with GATA3, and less than 50% with TFAP2C (49.12% and 45.52%, respectively), suggesting a stronger functional correlation between VGLL1, TEAD4 and GATA3 (Fig. 7a). In line with this, TE gene induction was impaired when *GATA3* was knocked down during the TELC induction (Supplementary Fig. 8k, l). Besides, we noticed a relatively lower co-binding between GATA3 and TFAP2C (37.98%), indicating that they bind to the VGLL1-TEAD4 complex both independently and in combination (Fig. 7a). A four-way Venn diagram comparison of the binding pattern of these four transcription factors showed the largest overlaps among VGLL1-TEAD4-GATA3-TFAP2C and VGLL1-TEAD4-GATA3 bound loci, whereas VGLL1-TEAD4-TFAP2C had a smaller overlap (Fig. 7b). Further investigation revealed that genes co-bound by all four factors were related to placenta development, cell growth regulation and histone modification (Fig. 7c), whereas genes co-bound by VGLL1-TEAD4-GATA3 were associated with multiple lineage development (Fig. 7d). These results suggest that the VGLL1-TEAD4 complex may function as an interacting hub for other human TE regulators including GATA3 and TFAP2C. Supporting these observations, co-immunoprecipitation showed interaction between these four factors in TELC-D5 cells (Supplementary Fig. 8m). Besides, the binding motifs of TEAD4, GATA3 and TFAP2C were all enriched in VGLL1-bound sites, TEAD4 and TFAP2C

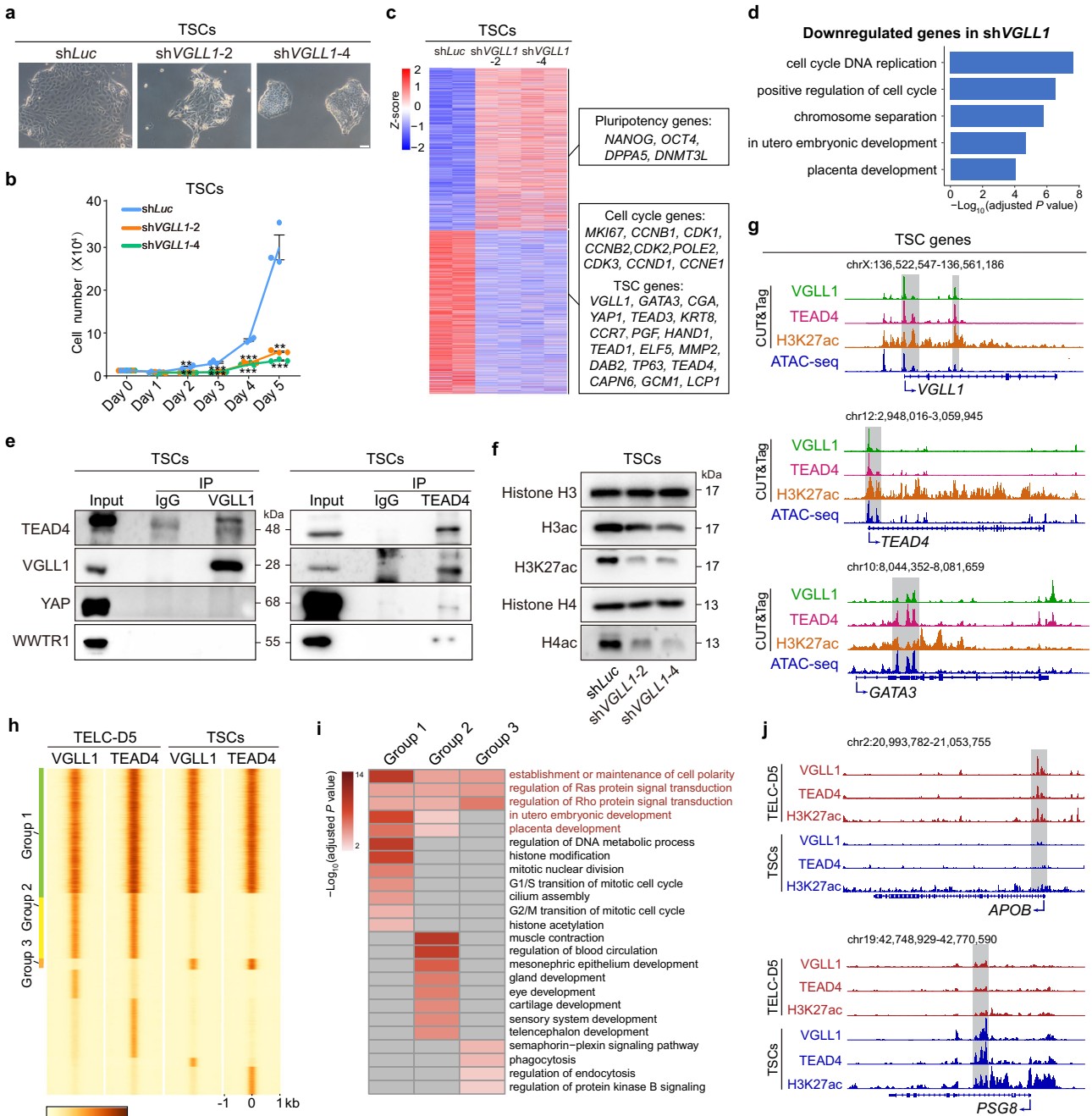

**Fig. 6 | VGLL1 is necessary for sustaining the human TSC identity. a** Phase contrast images of 4CL H9 ESC-derived TSCs transduced with sh*Luc*, sh*VGLL1*-2 or sh*VGLL1*-4. Scale bar, 100 μm. Representative of three independent experiments. **b** Analysis of cell numbers for 4CL H9 ESC-derived TSCs transduced with sh*Luc*, sh*VGLL1*-2 or sh*VGLL1*-4. Data are presented as the mean ± SEM, *n* = 3 biological replicates. *P* value was calculated using a two-tailed unpaired Student's *t*-test, ****P* < 0.001, ***P* < 0.01, **P* < 0.05. **c** Heatmap showing the expression of pluripotency, cell cycle and TSC genes for the indicated conditions. Example genes are shown for each cluster in the boxes. *n* = 2 biological replicates. **d** Enriched GO terms for downregulated genes in sh*VGLL1* compared to sh*Luc* control in 4CL H9 ESC-derived TSCs. *P* value was calculated using a hypergeometric test (one-sided) and adjusted for multiple testing using the Benjamini-Hochberg correction. **e** Immunoprecipitation using 4CL H9 ESC-derived TSC lysates with anti-VGLL1 (left panel) or anti-TEAD4 (right panel) and subsequent Western blotting with anti-TEAD4, anti-VGLL1, anti-YAP and anti-WWTR1. Representative of three independent

experiments. **f** Western blotting for the indicated histone marks for sh*VGLL1*-2 and sh*VGLL1*-4 compared to the sh*Luc* control in 4CL H9 ESC-derived TSCs. Representative of three independent experiments. **g** Genome browser tracks showing VGLL1, TEAD4, H3K27ac genomic enrichment peaks (CUT&Tag) and chromatin accessibility peaks (ATAC-seq) for representative TSC genes loci in 4CL H9 ESC-derived TSCs. **h** Co-occupancy analysis by signal density pileups of VGLL1, TEAD4 binding in 4CL H9 ESC-derived TELC-D5 cells and TSCs. The three major groups are: group 1 (common between TELC-D5 and TSC), group 2 (TELC-specific) and group 3 (TSC-specific). For clarity, other groups are not labelled. **i** Enriched GO terms for groups 1, 2 and 3 of panel **h**. *P* value was calculated using a hypergeometric test (one-sided) and adjusted for multiple testing using the Benjamini-Hochberg correction. **j** Genome browser tracks showing VGLL1, TEAD4 and H3K27ac genomic enrichment peaks (CUT&Tag) in 4CL H9 ESC-derived TELC-D5 cells and TSCs for representative gene loci from group 2 (upper panel) and group 3 (lower panel).

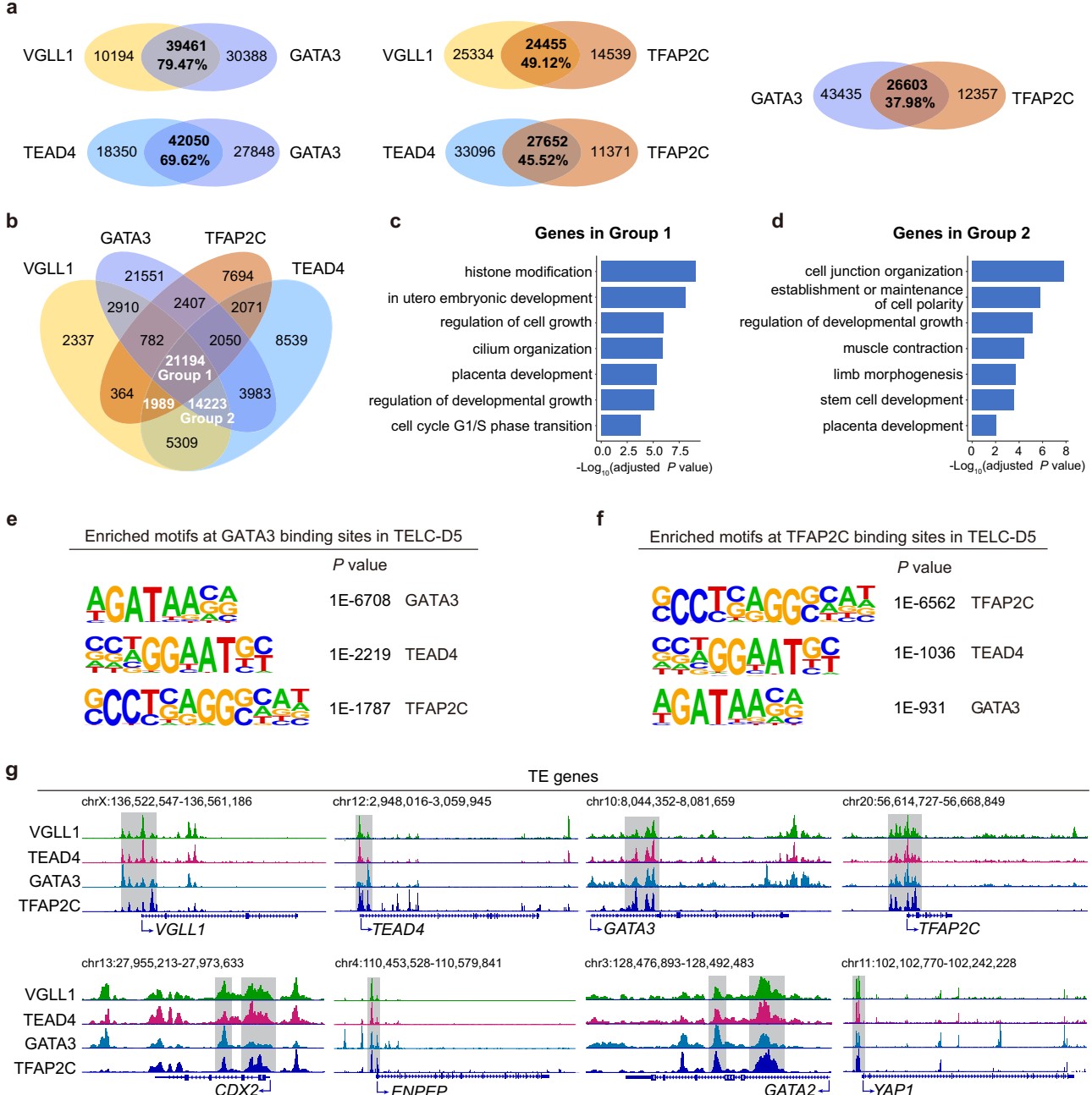

**Fig. 7 | VGLL1/TEAD4 cooperate with GATA3 and TFAP2C to regulate TE identity. a** Venn diagram showing the pairwise overlap between VGLL1-, TEAD4-, GATA3- and TFAP2C-bound sites in 4CL H9 ESC-derived TELC-D5 cells. **b** Four-way venn diagram showing the overlap between VGLL1-, TEAD4-, GATA3-, and TFAP2C-bound sites in 4CL H9 ESC-derived TELC-D5 cells. **c** Enriched GO terms for genes in Group 1 of panel **b**. *P* value was calculated using a hypergeometric test (one-sided) and adjusted for multiple testing using the Benjamini-Hochberg correction. **d** Enriched GO terms for genes in group 2 of panel **b**. *P* value was calculated using a hypergeometric test (one-sided) and adjusted for multiple testing using the

Benjamini-Hochberg correction. **e** Motif analysis using HOMER showing the significantly enriched DNA-binding motifs at GATA3-binding sites in 4CL H9 ESC-derived TELC-D5 cells. *P* value was calculated using a hypergeometric test (one-sided). **f** Motif analysis using HOMER showing the significantly enriched DNA-binding motifs at TFAP2C-binding sites in 4CL H9 ESC-derived TELC-D5 cells. *P* value was calculated using a hypergeometric test (one-sided). **g** Genome browser tracks showing VGLL1, TEAD4, GATA3 and TFAP2C binding in 4CL H9 ESC-derived TELC-D5 cells for representative TE loci.

motifs were enriched in GATA3-bound sites, and TEAD4 and GATA3 motifs were enriched in TFAP2C-bound sites (Fig. 4d and Fig. 7e, f). We also identified that VGLL1, TEAD4, GATA3 and/or TFAP2C co-bound genes were related to TE/TSC identity (Fig. 7g). Additionally, we observed binding of these factors at their own regulatory regions, pointing at positive feedback mechanisms. These results provide additional insights on the role of VGLL1 as a master regulator of human early placenta formation.

## Discussion

During human preimplantation embryonic development, early embryonic cells divide and differentiate to form a layer of epithelial cells, the TE, that consists of the outer cells of the blastocyst and later forms the placenta. The transiently formed TE lineage will develop into the cytotrophoblast, which is the stem cell niche of the placenta villi and gives rise to the multinucleated SCT and the invasive EVT. Failure to properly proceed with any of these events leads to

pregnancy loss or placental diseases[66]. Human cytotrophoblast identity can be captured in vitro from embryo blastocysts, first-trimester placenta or PSCs in the form of TSCs[15,16,19,21,27], which self-renew and can be differentiated into SCT and EVT. While the developmental trajectories are reasonably well understood, the key regulators and pathways underlying human TE formation and TSC maintenance are only beginning to be investigated and outstanding questions remain.

Using TELCs and TSCs derived from human naïve PSCs, we have demonstrated that VGLL1 functions as an essential regulator of their induction and maintenance. VGLL1 depletion in TELCs or TSCs affects an array of cell cycle-related genes, TE/placenta-related transcription factors and other regulators, which impairs TELC/TSC proliferation and cell identity. VGLL1 was known to be expressed in the placenta and be involved in preeclampsia[34,35] but the spatiotemporal kinetics of its expression during mammalian early development at the single-cell level had not been clarified. We show that VGLL1 expression substantially increases at the onset of both human and monkey TE specification, consistent with the idea that stage-specific regulators often play critical roles in controlling cell fate.

VGLL1 has been shown to be required for the induction of a panel of TE-related genes in primed human PSCs treated with BMP4[34], but this model does not recapitulate TE differentiation[26,27]. We show that VGLL1 cooperates with TEAD4 to regulate TE/TSC identity. Previous work had shown that VGLL1 interacts with TEAD4 to regulate gastric cancer malignancy[36] but a role for the VGLL1-TEAD4 complex in early human placenta formation had not been described. Interestingly, mouse TE formation in vivo requires nuclear translocation of YAP and the interaction with TEAD4, but VGLL1 is negligibly expressed at this stage[34]. In contrast, YAP is mostly cytoplasmic in human TE derived from naïve PSCs. In human first trimester placenta, YAP is located in both the nucleus and cytoplasm[13], suggesting that as development progresses the localization of YAP is dynamically regulated. Importantly, YAP/WWTR1 have been demonstrated to regulate human trophoblast cell identity[13,56]. Our findings suggest that YAP, WWTR1 and VGLL1 are critical partners of TEAD4 in TELCs and TSCs generated from human naïve PSCs. Further studies will be necessary to ascertain the potential crosstalk between them in the regulation of human TELCs/TSCs and the early placenta, and also the upstream signals controlling the subcellular distribution and function of both cofactors. Similarly, TEAD1 has recently been shown to regulate naïve PSC-derived TSC induction[30] and it is tempting to speculate that this transcription factor can also recruit VGLL1.

VGLL1 acts by promoting H3K27ac deposition at target loci, which in turn increases chromatin accessibility. This is consistent with the role of TEAD4 in promoting histone acetylation in cancer cells[67]. To fully understand this process, it will be important to determine what epigenetic modulators interact with VGLL1, which can for example be done by performing immunoprecipitation coupled with mass spectrometry. Furthermore, we show that the VGLL1-TEAD4 complex functions synergistically with other known master TE regulators such as the transcription factors GATA3 and TFAP2C[68]. The overall genome-wide binding overlap for all these factors is, however, only partial, pointing at both shared and unique functions. Importantly, some of these regulators are also enriched in naïve PSCs (e.g., TFAP2C, TEAD4) and amnion (e.g., GATA3), among other lineages. In the case of TEAD4, its binding is reorganized and expanded in TELCs compared to naïve PSCs, but some regions are shared. This suggests preparation for lineage switching when VGLL1 is recruited. Clarification of how these and other factors coordinate with each other to achieve their goals will help explain the continuum of cell fate decisions during early human development. An in-depth study of these mechanisms will not only provide new ways to understand normal and abnormal development but will also be instrumental to improve the quality and functionality of human PSC-derived embryo models[69–76].

## Methods

### Inclusion and ethics statement

The use of human H9 ESCs (WiCell, WA09), UH10 iPSCs and TSC^BT in this study is compliant with the 'Guidance of the Ministry of Science and Technology for the Review and Approval of Human Genetic Resources' and was approved by the 'Life Science and Medical Ethics Committee' of the Guangzhou Institutes of Biomedicine and Health under license number GIBH-LMEC2023-001-01(AL).

### Cell culture

HEK293T cells and ICR mouse embryonic fibroblast-derived feeders were maintained in DMEM (Corning) supplemented with 10% fetal bovine serum (NATOCOR). Primed human PSCs, including H9 ESCs and UH10 iPSCs[50,51] were routinely cultured in mTeSR™1 medium (STEMCELL). For passaging, primed PSCs were washed with DPBS (Hyclone) once and treated with 0.5 mM EDTA (Invitrogen, 15575020) for 5 minutes. Then, EDTA was removed and cells were passaged as small clumps using a Pasteur pipette. Human TSC^BT were cultured in TSC medium[15] (DMEM/F12 supplemented with N2 and B27, penicillin-streptomycin, Glutamax, β-mercaptoethanol, 1.5 µg/ml L-ascorbic acid, 50 ng/ml EGF (PeproTech), 0.5 µM A83-01 (Selleck), 1 µM SB431542 (Selleck), 2 µM CHIR99021 (Axon), 0.8 mM VPA (Vetec) and 5 µM Y-27632 (Axon) supplemented with 10 µM Y-27632. Human H9 ESCs were purchased from WiCell Research Institute, human UH10 iPSCs were provided by Dr. Guangjin Pan (Guangzhou Institutes of Biomedicine and Health, Chinese Academy of Sciences, China), and TSC^BT were provided by Dr. Hiroaki Okae and Dr. Takahiro Arima (Department of Informative Genetics, Tohoku University Graduate School of Medicine). All cell lines were negative for mycoplasma.

### Naïve PSC induction

Generation of 4CL human naïve PSCs was performed as reported[33]. Briefly, primed PSCs were dissociated into single cells with a 1:1 mixture of 0.5 mM EDTA and TrypLE Express (Gibco) and plated at a density of 100,000 to 150,000 cell/well of 6-well plate on feeders in mTeSR™1 medium supplemented with 10 µM Y-27632 (Axon, 1683). Twenty-four hours later, the medium was switched to 4CL medium[33], which was refreshed daily. Cells were passaged as single cells (1:5 to 1:8) every 3 to 4 days. PXGL and HENSM naïve PSCs were induced as previously reported[52,53].

### TE and TSC induction

TE and TSC induction were performed as previously described[15,26]. Briefly, for generating TE, human naïve PSCs were dissociated into single cells with a 1:1 mixture of 0.5 mM EDTA and TrypLE Express and plated at a density of 100,000 cell/well of 6-well plate on Geltrex in TE induction medium (1:1 mix of Neurobasal medium (Gibco) and DMEM/F12 (HyClone) supplemented with N2 (Gibco) and B27 (Gibco), penicillin-streptomycin (HyClone), Glutamax (Gibco), β-mercaptoethanol (Sigma), 1 µM PD0325901 (Axon) and 1 µM A83-01 (Selleck)) for 5 days. TELCs at day 5 were used for downstream analysis throughout this study. For generating expandable TSCs, TELCs at day 5 were dissociated into single cells with TrypLE and plated on Geltrex-coated 6-well plates at a 1:4-1:8 split ratio in TSC medium (DMEM/F12 supplemented with N2 and B27, penicillin-streptomycin, Glutamax, β-mercaptoethanol, 1.5 µg/ml L-ascorbic acid, 50 ng/ml EGF (PeproTech), 0.5 µM A83-01 (Selleck), 1 µM SB431542 (Selleck), 2 µM CHIR99021 (Axon), 0.8 mM VPA (Vetec) and 5 µM Y-27632 (Axon) supplemented with 10 µM Y-27632.

### EVT and SCT differentiation

Differentiation of TSCs toward EVT and SCT were performed as previously described[60]. Briefly, for EVT differentiation, TSCs were plated at a density of 70,000 cell/well on Geltrex-coated 6-well plates in EVT medium (TeSR™-E8 medium (STEM CELL), 2.5 µM SB431542 and 2.5 ng/ml EGF). Over the course of 12 days, medium was refreshed every

2 days. For SCT differentiation, TSCs were plated at a density of 40,000 cell/per well on Geltrex-coated 6-well plates in SCT medium (TeSR™-E6 medium (STEM CELL), 20 ng/ml activin A and 50 ng/ml EGF). Medium was refreshed every two days during the course of 14 days.

## shRNA mediated knockdown
For shRNA-mediated *VGLL1*, *TEAD4*, *YAP1*, *WWTR1*, *TP53* and *GATA3* knockdown experiments, shRNA inserts were cloned into pLKO.1 vector. Lentiviruses were generated with HEK293T cells using Lipofectamine 3000 (Invitrogen) following the manufacturer's instructions. One day after passaging, human naïve PSCs, cells at day 1 of TELC-induction (for *TP53* knockdown) or TSCs were infected with shRNA viruses for 6-8 hours. Transduced cells were subjected to puromycin selection (1 µg/mL) 36 hours later and continued for 1-2 days until control cells died. Transduced human naïve PSCs were used for TELC differentiation as previously described. Transduced TSCs were collected without further passaging or used for further EVT and SCT differentiation. All shRNA target sequences are listed in Supplementary Table 1.

## Generation of knockout cells
Single guide RNAs (sgRNAs) targeting *VGLL1* and *TEAD4* exons were cloned into lentiCRISPR-v2 plasmids (Addgene, 52961) for KO cell line generation. H9 ESCs were transfected with sgRNA-coding plasmids using Lipofectamine 3000 (ThermoFisher) following the manufacturer's instructions. Puromycin selection (1 µg/mL) was conducted 2 days post-transfection and continued for 1-2 days until control cells were dead. Single-cell derived clones were picked for further expansion and PCR-based genotyping was performed to test frameshift mutations. Homozygous KO were further confirmed by TA cloning followed by Sanger sequencing, and also by Western blotting. All sgRNA sequences are listed in Supplementary Table 2. Primers used for genomic PCR are listed in Supplementary Table 3.

## Growth curve generation
A total of $1.25 \times 10^4$ TELCs or TSCs were seeded at day 0 per well of a 12-well plate, or a total of $8 \times 10^4$ TELCs or $2.5 \times 10^4$ TSCs were seeded at day 0 per well of a 6-well plate. The medium was changed every day. The cell number was counted for 3 wells as triplicates using a hemocytometer. Three independent experiments were performed.

## Immunofluorescence, Western blotting and immunoprecipitation
For immunofluorescence, cells were washed with PBS and then fixed with 4% paraformaldehyde for 10 minutes at room temperature and permeabilized with 3% BSA (bovine serum albumin) +0.2% Triton X-100 in PBS for 60 minutes at room temperature. Cells were incubated with primary antibodies overnight at 4 °C and subsequently with secondary antibodies for 1 hour at room temperature. Finally, cells were stained with DAPI for 5 minutes and observed with a confocal microscope. Immunofluorescence was assessed using a Leica TCS SP2 spectral confocal microscope. For Western blotting, cells were washed with PBS and lysed on ice in RIPA buffer (50 mM Tris-HCl (pH 7.5), 150 mM NaCl, 0.25% sodium deoxycholate, 0.1% NP-40 and 0.1% Triton X-100) supplemented with a proteinase inhibitor (Cocktail, Roche). Samples were sonicated and subjected to SDS–PAGE and transferred onto a PVDF membrane (Millipore). Membranes were blocked with 5% non-fat milk in TBST and then sequentially incubated with primary antibody overnight at 4 °C and secondary antibodies for 1 hour at room temperature. Signals were detected by Amersham ECL (GE Healthcare) and visualized with a FUSION SOLO 4 M machine (Vilber Lourmat) and analyzed with FusionCapt Advance Solo 4.16.15. For nuclear-cytoplasmic fractionation, cells were scraped down from the culture plates and lysed in 150 µL lysis buffer (PBS containing 0.04% NP-40 and proteinase inhibitor) on ice with occasional pipetting. 50 µL of sample were kept as whole cell lysate input. The remaining samples were centrifuged and 50 µL supernatant was kept as a cytoplasmic fraction. The precipitated nuclear fraction was washed four times and sonicated. The whole cell lysate input, cytoplasmic fraction and nuclear fraction were used for SDS-PAGE analysis. For immunoprecipitation, cells were lysed in TNE lysis buffer (50 mM Tris-HCL (pH 7.5), 250 mM NaCl, 0.5% NP-40 and 1 mM EDTA) containing a proteinase inhibitor cocktail and rotated on ice for 15 minutes. Lysates were homogenized by a 0.4 mm syringe needle and then centrifuged at 13,000 g for 15 minutes at 4 °C. Supernatants were pre-cleaned with prewashed protein A/G for 1 hour at 4 °C. Same volume of TNEG buffer (50 mM tris-HCl (pH 7.5), 50 mM NaCl, 0.5% NP-40, 20% glycerol, and 1 mM EDTA) was added to dilute the supernatants. Primary antibodies were incubated with the diluted supernatants overnight at 4 °C and then incubated with prewashed protein A/G magnetic beads for 3 hours at 4 °C. Samples were then loaded on an Invitrogen magnetic separator and beads were washed four times with washing buffer (20 mM Tris-HCl pH 7.6 and 140 mM NaCl). Proteins were eluted with 1x loading buffer by boiling at 80 °C in a water bath for 10 minutes and subjected to SDS-PAGE analysis. Antibodies used for immunofluorescence, Western blotting and immunoprecipitation are listed in Supplementary Table 5. Unprocessed scans of Western blotting for all figures are provided in Supplementary Fig. 9.

## RNA isolation, RT–qPCR and RNA-seq library preparation
Total RNA was isolated using RNAzol (MRC) based on the manufacturer's instructions. RT-qPCR analysis was performed using the SYBR Premix ExTaq Kit (Takara, RR420A) with an ABI 7500 real-time PCR machine. All RT-qPCR primers are listed in Supplementary Table 4. Bulk RNA-seq library preparation and sequencing were performed at Berry Genomics Co. Ltd. (Beijing, China).

## ATAC-seq and CUT&Tag library construction
For ATAC-seq library preparation, 50,000 cells were washed once with cold PBS and resuspended in 50 µl of lysis buffer (10 mM Tris-HCl (pH 7.4) 10 mM NaCl, 3 mM MgCl2, and 0.1% (v/v) IGEPAL CA-630 (Sigma)). The suspension was then centrifuged at $500\,g$ for 10 minutes at 4 °C, followed by the addition of 50 µl of the transposition reaction mixture of the TruePrep DNA Library Prep Kit (Vazyme, TD502). Samples were then incubated at 37 °C for 30 minutes. Transposition reactions were cleaned up using the MinElute PCR Purification Kit (Qiagen, 28004). ATAC-seq libraries were subjected to five cycles for preamplification and amplified by PCR for an appropriate number of cycles. Amplified libraries were purified using the QIAquick PCR Purification Kit (Qiagen, 28104). Library concentration was measured using the VAHTSTM Library Quantification Kit (Vazyme, NQ101). For CUT&Tag library construction, Hyperactive In-Situ ChIP Library Prep Kit for Illumina (Vazyme, TD902-01) was used. In brief, 100,000 cells were incubated with Concanavalin A-coated magnetic beads and then primary antibodies were added to incubate overnight at 4 °C. Tubes were placed on a magnetic rack to remove the supernatants. Secondary antibodies were added and samples were incubated at room temperature for 1 hour. After placing the tubes on the magnetic rack and removing the supernatants, hyperactive pG-Tn5/pA-Tn5 transposon (incubated at room temperature for 1 hour) were added. Tagmentation was performed for 1 hour and DNA was extracted using phenol–chloroform–isoamyl alcohol. Purified DNA fragments were amplified by PCR for 15–20 cycles and the PCR products were purified following the kit's instructions. CUT&Tag library construction were performed in duplicate. Libraries were sequenced by Berry Genomics Co. Ltd. (Beijing, China). Primary antibodies used for CUT&Tag library construction are listed in Supplementary Table 5.

## Bulk RNA-seq, ATAC-seq and CUT&Tag data analysis

For bulk RNA-seq, reads were trimmed of adapters and low-quality reads were removed by fastp[77] (v0.21.0) (https://github.com/OpenGene/fastp) with the default option. Trimmed data were aligned to the Ensembl v76 (hg38) transcript annotations using STAR[78] (v2.7.3a) with the settings *--outSAMtype BAM* Unsorted, *--quantMode TranscriptomeSAM*, *--outSAMheaderHD\@*HD VN:1.4 SO:unsorted. Gene counts were calculated using RSEM (v1.2.18) with the settings *--paired-end*. After removing the genes with zero count in all samples, DEGs were determined using DESeq2[79] (v1.24) in R (v3.6.0). Heatmaps were generated using pheatmap (v1.0.12) in R (v3.6.0). Volcano plots were generated from DESeq2 output files using EnhancedVolcano (v1.8.0) in R (v3.6.0) with the settings *pCutoff* = 10e-3, *FCcutoff* = 1.0. Processing of the public bulk RNA-seq data was the same as that of our bulk RNA-seq data. For ATAC-seq and CUT&Tag, reads were trimmed of adapters and low-quality reads were removed by fastp[77] (v0.21.0) (https://github.com/OpenGene/fastp) with the settings *-g*, *--length_required* = 10, *--n_base_limit* = 5. Trimmed data were first aligned to the hg38 assembly using Bowtie2 (v2.2.5) with the settings *--very-sensitive*. Low quality mapped reads were removed using Samtools with the settings *-q* 30. Duplicated reads were collapsed using Picard (v1.9.0). Peaks were called using MACS2 (v2.1.0) with the settings *-B*, *--nomodel*, *--keep-dup* 1, *-g* hs, *--call-summits*, *-q* 1e-05 and then were annotated using ChIPseeker[80] (v1.26.2) *annotatePeak* tools. Pie charts of genomic annotation were generated using ggplot2 (v3.3.3) in R (v3.6.0). Motif discovery was performed using HOMER (v4.9.1) findMotifsGenome.pl tools with the default option. Venn diagrams were generated from MACS2 narrowPeak files using ChIPpeakAnno[81,82] (v3.24.2) *findOverlapsOfPeaks* and *makeVennDiagram* tools. BigWig files were generated using MACS2 (v2.1.0) bdgcmp tools with the settings -m logLR, -p 0.00001 and deeptools(v3.4.3) *bamCoverage* tools with the settings *−ignoreDuplicates*, *--normalizeUsing* CPM, *--minMappingQuality* 30. Co-occupancy analysis by tag density pileups was done by deeptools (v3.4.3) *computeMatrix reference-point* tools with the settings *-skipZeros*, *plotHeatmap* tools with the settings *--sortRegions* keep and *plotProfile* tools with the settings *−perGroup*, *--kmeans* 1. Functional annotation was performed by ClusterProfiler[83] (v3.6.0) *enrichGO* tools in R (v3.6.0) and barplots were generated using ggplot2 (v3.3.3) in R (v3.6.0).

## Library preparation for scRNA-seq

ScRNA-seq libraries were prepared using a DNBelab C Series Single-Cell Library Prep set (MGI, 1000021082)[84]. Briefly, cells were dissociated into single cells with a 1:1 mixture of 0.5 mM EDTA and TrypLE Express (Gibco). After passing through 70 μM cell strainers, cell suspensions were converted to barcoded scRNA-seq libraries according to the manufacturer's protocol. Qubit ssDNA Assay kit (Thermo Fisher Scientific, Q10212) was used to quantify the sequencing library concentration. The resulting libraries were sequenced using a DIPSEQ T1 sequencer at the China National GeneBank (CNGB) in Shenzhen.

## scRNA-seq data analysis

**Raw data processing.** Raw sequencing reads were filtered and demultiplexed using PISA (v0.2) (https://github.com/shiquan/PISA). Filtered reads were aligned to hg38 genome reference using STAR[78] (v2.7.3a) and sorted by sambamba (v0.7.0)[85]. Cell versus gene UMIs (Unique Molecular Identifiers) count matrices were generated by PISA.

**Quality control.** Count matrices were processed using the Seurat package (v3.1.4)[86] in R (v3.6.0). Genes expressed in less than three cells were filtered out. Cells with less than 15% mitochondrial reads, less than 500 genes and 3,000 UMIs detected were excluded from further analysis. Doublet removal was performed by DoubletFinder[87] with the default parameter. Public data were re-analyzed from E-MTAB-3929,

PRJEB30442, GSE136447, GSE74767, GSE84892 and CNP0001454. Quality control for public datasets was processed according to their respective reference articles.

**Analysis of scRNA-seq data.** Seurat package[88] (v3.1.4) in R (v3.6.0) was used to create Seurat objects for analysis. Public datasets were annotated using the gene lists provided in their original articles. We first used the Seurat *FindIntegrationAnchors* function to integrate our scRNA-seq data of TELC-D5 with human embryo public scRNA-seq data (E-MTAB-3929, PRJEB30442, GSE136447)[40–42]. Four Seurat objects were merged and then three thousand genes that were repeatedly variable across datasets were selected by *SelectIntegrationFeatures* tools. *FindIntegrationAnchors* was then performed with the settings *reduction*=rpca. All datasets were finally integrated by *IntegrateData* with default parameters. Principal components were selected by principal component analysis (PCA). UMAP was then used for visualization. Plots were generated using *DimPlot* and *DotPlot*. For scRNA-seq data of TELC-D5 cells, Seurat object was processed using *NormalizeData*, *FindVariableFeatures*, *ScaleData*, *RunPCA* and *RunUMAP* functions in turn. Dimension reduction started with PCA on 3,000 significantly variable genes and the first thirty principal components were selected as input for UMAP visualization. Correlation analysis was based on the 10,000 significantly variable genes and cor () function in R (v3.6.2). Correlation heatmap were generated by pheatmap (v1.0.12). UMAP was used for visualization of public embryo scRNA-seq data (E-MTAB-3929, GSE74767, GSE84892, PRJEB30442). The single-cell trajectory was analyzed using the count matrix of public embryo scRNA-seq data by Monocle 2 (v4.1.0). DEGs between cell types were identified using *FindAllMarkers* function of Seurat package with the settings *only.pos* = TRUE, *min.pct* = 0.25, *logfc.threshold* = 0.25 for trajectory analysis. Dimensions were reduced by using the *reduceDimension* function with the method of DDRTree. Lineage trajectory and branch points were then constructed. Pre-lineage of the trajectory was chosen as the root. Trajectory was plotted by *plot_cell_trajectory* function of Monocle 2 (v4.1.0). Expression patterns of key landmark genes along the trajectory were generated by *plot_genes_branched_pseudotime* function of Monocle 2. Plots were generated using stacked violin functions.

## Statistics and reproducibility

Data of bar charts are represented as mean ± standard error of the mean (SEM). Significance was tested using a two-tailed unpaired Student's t-test. The related $P$ values or $P$ value ranges are shown in the figure legend. In all cases, $*P < 0.05$, $**P < 0.01$, $***P < 0.001$. The number of replicates for each experiment is presented in the figure legends.

## Reporting summary

Further information on research design is available in the Nature Portfolio Reporting Summary linked to this article.

# Data availability

All bulk sequencing data (RNA-seq, ATAC-seq and CUT&Tag) and single-cell sequencing data (scRNA-seq) reported in this paper are deposited in the Gene Expression Omnibus (GEO) database under the accession number GSE193621. The raw single-cell sequencing data are also deposited in the CNGB Nucleotide Sequence Archive under the accession number CNP0004944. Public scRNA-seq dataset of human early embryo development were downloaded from EBI ArrayExpress (E-MTAB-3929)[40], ENA database (PRJEB30442)[41] and GEO database (GSE136447)[42]. scRNA-seq datasets of monkey and mouse early embryo development were downloaded from GEO database (GSE74767[48], GSE84892[49]). Bulk RNA-seq and scRNA-seq dataset of 4CL H9 naïve ESCs were downloaded from CNGBdb (CNP0001454)[33]. Source data are provided with this paper.

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

## Acknowledgements

We thank all members of the Laboratory of Integrative Biology of the Guangzhou Institutes of Biomedicine and Health ang BGI research for their support. We thank the instrument platform and the super-computing center of the Guangzhou Institutes of Biomedicine and Health for their technical support. We thank the CNGB for providing technical support. This work was supported by the National Natural Science Foundation of China (32370848, 32250710149 and U20A2015 to M.A.E., 32150410348 to M.A.M. and 32200669 to W.L.), the Guangdong Basic and Applied Basic Research Foundation (2021B1515120075 to M.A.E.), the Science and Technology Planning Project of Guangdong Province (2023B1212060050 to W.L.), the Guangzhou Science and Technology Foundation (2024A04J4702 and 202102020176 to W.L.), the CAS President's International Fellowship Initiative (CAS-PIFI) for special experts (2020FSB0002 to M.A.M.) and the Foreign Expert Project-Young Talent Program (QN2022031001L to M.A.M.).

## Author contributions

W.L., M.A.M., and M.A.E. conceived the original idea and supervised this study. Z.L. contributed to the original idea. Y.Y. and W.J. conducted most of the experiments with help from Z.L., Y.Li, Hao L., L.F., Y.J., J.Lai., Haiwei

L., B.J.S., Y.Z., Y.Lv., Y.S. and C.L. W.J. performed most of the bioinformatics analysis with help from J. Li, L.W. and Y.Lai. W.L., M.A.M., and M.A.E. interpreted the data and wrote the manuscript with input from W.J., A.P.H., T.Z. and L.L.

## Competing interests

The authors declare no competing interests.
