## [Peer Review File · Nature Communications]

VGLL1 cooperates with TEAD4 to control human trophectoderm lineage specificationREVIEWER COMMENTS

Reviewer #1 (Remarks to the Author):

Wenjuan Li and colleagues investigate the role of the transcriptional co-factor VGLL1 during trophoctoderm (TE) lineage specification from human naïve pluripotent stem cells (PSCs). The factors regulating human TE specification remain poorly understood, but recently developed methods for deriving TE and trophoblast stem cells (TSCs) from naïve PSCs offer new model systems to gain insights into mechanisms of early placental development. The authors use human naïve PSCs generated with their recently developed 4CL culture medium to produce TE and TSCs. Based on shRNA knockdowns, they show that VGLL1 is required to generate TE from human naïve PSCs. They also map the genome-wide binding sites of VGLL1 by CUT&Tag analysis, which reveals enrichment of the TEAD4 DNA binding motif at VGLL1-bound sites. This leads them to hypothesize that VGLL1 interacts with TEAD4 to promote human TE specification. Indeed, TEAD1 and VGLL1 interact with each other based on reciprocal co-IP experiments. A role for VGLL1 in regulating H3K27ac deposition is inferred from a global reduction in H3K27ac levels upon VGLL1 knockdown and the strong enrichment of H3K27ac at VGLL1-TEAD4 co-bound sites. Finally, the authors show that VGLL1 is also required to sustain a trophoblast identity in expandable TSCs.

This is an interesting manuscript that sheds new light on the mechanisms regulating human TE specification. Strengths of this paper are the use of human naïve PSCs to investigate TE specification and the computational analyses of VGLL1 expression in human, monkey and mouse embryos. However, several additional experiments and analyses are required to better support the main conclusions.

Specific points:

1. The main conclusions rely on shRNAs to knock down VGLL1 expression in TE (Fig. 3 and 5) and TSCs (Fig. 6). While the hairpins achieve good knockdown efficiency, a general concern with this approach is the potential for off-target effects. Since VGLL1 is not expressed in pluripotent cells, it should be possible to knock out this gene in naïve PSCs and demonstrate that TE specification is affected using a clean homozygous deletion.
2. The authors use the recently reported “4CL” conditions for naïve PSCs (Mazid et al., 2022). These conditions were only published very recently and by the same lab. To ensure the broad relevance of this work, it would be helpful to reproduce at least some of the key findings using TE cells that were generated from naïve PSCs in more widely established culture conditions, such as PXGL or 5iLA.
3. Related to the previous point, it would be helpful if the authors could demonstrate using bioinformatic integration that their d5 TE cells in fact cluster more closely with pre-implantation TE than post-implantation CTBs. They cite the work from Io et al. and Guo et al., but those studies used naïve PSCs derived under different conditions. In their prior study (Mazid et al.), the authors directly converted 4CL naïve cells into TSCs and did not generate pre-implantation TE cells.

4. In Fig. 2a-d the human embryo data from Petropoulos et al. (2016) are used to examine the expression of VGLL1 and other TE regulators. This is a good starting point, but I think it would be instructive to look at another human embryo single cell dataset as well, for example Meistermann et al., Cell Reports, 2021. A potential concern about the Petropoulos data is that a subset of their EPI and PE cells likely originate from TE (Stirparo et al., 2018).

5. In Fig. 3C and Fig. 6a it would be helpful to use higher magnification images to see the impact of VGLL1 knockdown on TE and TSC morphology.

6. Motif analysis at VGLL1 binding sites in d5 TE reveals enrichment of the TEAD4 motif (Fig. 4d). Is this analysis specific for TEAD4 and not its paralogs, TEAD1 and TEAD3? The latter two are also expressed in human trophoblast cells and may have overlapping functions.

7. TEAD4 and VGLL1 interact with each other based on co-IP assays (Fig. 4f) and co-occupy target genes in d5 TE cells (Fig. 4h-i). Based on these data, the authors conclude that “cooperation between VGLL1 and TEAD4 is necessary for human TE induction from naïve PSCs” (line 223-224). Additional evidence is needed to support this assertion. For example, does VGLL1 knockdown – or knockout - impair the ability of TEAD4 to bind its target genes?

8. Global levels of H3K27ac decline upon VGLL1 knockdown (Fig. 5a) and H3K27ac is enriched at VGLL1-bound sites. This leads the authors to conclude that VGLL1 is involved in opening chromatin at target loci through increased histone acetylation. It would be more convincing if the authors could include ATAC-seq and H3K27ac CUT&Tag analysis on VGLL1 knockdown – or, ideally, knockout – cells, showing a specific depletion of open chromatin at VGLL1 target sites.

9. Fig. 6 shows that VGLL1 is required not only in pre-implantation TE, but also in expandable TSCs that resemble post-implantation CTBs. Since TSCs derived from naïve PSCs have some different epigenetic properties from primary TSCs, can the authors reproduce these findings using a TSC line that was derived directly from blastocysts or placental tissues?

10. In Fig. 7 the authors integrate the CUT&Tag data for VGLL1 and TEAD4 with previously published ChIP-seq data for GATA3 and TFAP2C to argue that all four factors cooperate in the human TE regulatory network. Aside from technical differences related to the comparison of CUT&Tag vs. ChIP-seq data, the GATA3 and TFAP2C data were generated using primed human PSCs treated with BMP4 (Krendl et al., 2017). As the authors themselves point out, this model does not recapitulate TE differentiation (line 367). Therefore, I’m confused why they rely on these previously published ChIP-seq data from the BMP4 model. It would be better to perform CUT&Tag for GATA3 and TFAP2C in the same d5 naïve TE cells that are analyzed throughout this manuscript.

Reviewer #2 (Remarks to the Author):

In this work, Yueli Yang and colleagues sought to investigate the role of VGLL1 in early human trophectoderm (TE) development by employing recently developed model of naïve pluripotent cells to TE-like state transdifferentiation. They showed that VGLL1 is expressed at high level in human TE cells derived from naïve PSC, as well as in the TE compartment of human and monkey blastocysts. As reported before (PMID: 29361559), the authors demonstrated the interaction between VGLL1 and TEAD4, and its importance in early TE development. Using Cut and Tag, ATAC-seq, RNA-seq and ChIP-seq, they show that many regions occupied by VGLL1 occupied as well by TEAD4. By knocking down either VGLL1 or TEAD4 they show that the conversion to TE-like state is impaired mainly due to cell cycle arrest and reduced TE-specific gene expression. This notion was also true for TSC-derived from the same transdifferentiation model. Finally, they show that GATA3 and TFAP2C also occupy a large fraction of these regions but work independently. Overall, this is a very nice study, and the majority of the experiments and results are of high quality and convincing. However, some issue should be addressed prior to publication.

Major comments:

1. One main concern of the reviewer is the claim regarding the importance of VGLL1 in inducing the TE-like state. The reviewer is convinced that the cell cycle is impaired in the absence of VGLL1 or TEAD4 but is concerned whether the entire effect seen in the knockdown (KD) cells is related to cell cycle solely. First, the reduction of TE-specific genes is relatively mild in the KD cells (CGB and CGA are not TE-specific markers and are expressed in differentiated cells as well) and the morphology of the KD cells seems similar to the reviewer as to the control, albeit the colonies are definitely smaller. To exclude the possibility that the effect seen is only due to cell cycle arrest, the reviewer suggests to KO/KD p53 in the naïve pluripotency VGLL1 and TEAD KD cells and repeat the experiment. Showing normal proliferating cells and reduced expression of TE-specific genes will validate the authors' claim.
2. Is there a reason why the authors used shRNAs and not CRISPR/Cas9 mediated KO? ShRNAs known to have non-specific effect especially on proliferation so if possible, the authors should repeat at least one main assay with KO cells.
3. To support the authors' claim that VGLL1 and TEAD KD cells are indeed impaired, a functional assay should be conducted. This can be shown for example by directed differentiation toward STs and EVT.
4. The authors show that TE-like and TSC VGLL1 KD cells still express pluripotency genes (Figs. 3e and 6c). While it could be reasonable for the TE-like cells the reviewer is confused as to how this can happen in TSCs? Are the authors claim that VGLL1 constantly suppresses pluripotency genes at the TSC stage? The authors should show the expression of some of those pluripotency genes at the protein level (staining or western blot) and explain this result.
5. Given the high overlap between GATA3 and TFAP2C to VGLL1 regions, the reviewer believe it is important to show whether they are working as a complex by performing CO-IP experiments.
6. It is not clear how the authors explain the results shown in Fig. 6j? The regions (group 2) that separate the role of VGLL1 in TE vs TSCs are not relevant to the TE state at all. How do the authors explain enrichment for muscle contraction, blood circulation and gland and eye development in these cells? Could it be that these results actually demonstrate a non-homogenous differentiation toward the TSC state at day 5 of differentiation? Could it be that TE-like cells in this protocol contain also other differentiated cells? from different cell types?

Minor comments:

7. Fig. 1c- the reviewer think it is important to show that the examined genes are not expressed in primed ESCs. Please include primed ESCs to the staining panel.

8. Since no study rigorously demonstrated that following 5 days of differentiation the cells are bona fide TE cells, the authors should change the term "TE" to "TE-like" throughout the entire manuscript.

9. Fig. 2d, 2h- Title is missing for the Y axis.

10. Fig. 3f- more TE/TSC-specific genes should be examined such as GATA2, TFAP2C, TFAP2A, KLF5 etc.

11. The authors conducted ATAC-seq and examined loci co-bound by VGLL1 and TEAD. However, they did not analyze loci bound only by VGLL1 or TEAD4. Can this be added to the manuscript.

12. It has been suggested lately that GATA3 binds to VGLL1 promoter in trophoblast progenitor and increases its expression compared to naïve ESCs (PMID: 35637409). Here, the authors claim that VGLL1 is also a regulator of GATA3 at the same developmental stage. Can the authors test GATA3 KD transdifferentiation experiment and compare it to VGLL1 KD differentiation experiment, in at least one main assay.

Reviewer #3 (Remarks to the Author):

In this report, the authors define a role for VGLL1 as a co-factor for TEAD4 in the regulation of the trophoblast (TE) and trophoblast stem (TS) cell states. The authors provide a breadth of experimental work to determine expression profiles and loss-of-function experiments on the TE and TS cell states. In addition, the authors examine genome-wide distributions of VGLL1 and TEAD4 in order to identify direct targets of TEAD4 and VGGL1 action. The role of VGLL1 as a regulator of TEAD4 action was contrasted to roles for YAP and WWTR1 (TAZ) in TEAD4 action. Furthermore, species specific biology of TE and TS cell stem states were investigated. The manuscript is well-written and addresses an important biological problem. The work represents a significant contribution to understanding regulatory mechanisms controlling trophoblast lineage development; however, some issues with the experimentation and interpretation of data are evident. Specific concerns are elaborated below.

1) The experimentation was performed with the H9 human embryonic stem (ES) cell line. These cells were used to create naïve pluripotent stem cells and then these naïve pluripotent stem cells were converted to trophoblast (TE) cells and to trophoblast stem (TS) cells. The role of VGLL1 was investigated in these cell models derived from H9 human ES cells. Not all pluripotent stem cells exhibit similar abilities to differentiate into trophoblast cell lineages (Cinkornpumin et al. Stem Cell Reports. 2020;15(1):198-213, PMC7363941). Thus, it is important to demonstrate that the phenomena and mechanisms described in this report are not unique to derivatives of H9 ES cells. Do H9 ES cell-derived TS cells possess the capacity to differentiate into extravillous trophoblast and syncytiotrophoblast? Utilization of cytotrophoblast-derived TS cells as a reference standard for some of the work would be very helpful.

2) The findings published in this report emphasize the importance of VGLL1 in serving as a cofactor for

TEAD action in human TE and TS cells and suggest that YAP1 or WWTR1 are not serving as TEAD cofactors in these cell lineages. There are published reports implicating YAP1 and WWTR1 as cofactors for TEAD family transcription factors in maintenance of the human TS cell/cytotrophoblast stem state (Meinhardt et al. Proc Natl Acad Sci U S A. 2020 117(24):13562-13570. PMC7306800; Saha et al. Proc Natl Acad Sci U S A. 2020; 117(30):17864-17875. PMC7395512; Ray et al. Proc Natl Acad Sci U S A. 2022; 119(36):e2204069119. PMC9457323). These reports directly utilized trophoblast cell populations isolated from placenta and verified aspects of their work with trophoblast tissues. Is there a difference in roles for VGLL1, YAP1, and WWTR1 as TEAD cofactors based on the in vitro model system utilized? Are the authors results unique to TE and TS cells derived from H9 ES cells? Are actions for YAP1 and WWTR1 in TS cells published by others compatible with the authors observations? The authors need to address these issues.

3) The authors should consider discussing the evolutionary significance of utilizing different TEAD cofactors in mouse versus primate trophoblast cells. Is there an advantages in using distinct cofactors for mouse versus primate trophoblast cells? Does each cofactor direct TEAD action to unique parts of the mouse versus primate genome?

4) In the last set of data discussed in the Results section, the authors compare their findings with datasets generated from Krendl et al. 2017. These datasets from Krendl et al. were generated from BMP4-treated human pluripotent stem cells. As stated in the report, this population of cells exhibits similarities to amnion and not trophoblast. Thus, the merits of using these BMP4-treated human pluripotent stem cell datasets need to be better justified. Reliance exclusively on the expression of APA in a cell population could be misleading.

Reply to the reviewers' comments

We thank all three reviewers for their constructive comments and the overall positive assessment of our manuscript. Based on the reviewers' questions and suggestions, we have now performed new experiments and analyses and incorporated them together with textual modifications in the revised manuscript. A point-by-point response to the reviewers' comments is provided below. Major changes in the manuscript to address these comments are highlighted in red. We hope that the reviewers will be satisfied with our revised manuscript and responses.

Reviewer #1 (Remarks to the Author):

“Wenjuan Li and colleagues investigate the role of the transcriptional co-factor VGLL1 during trophoctoderm (TE) lineage specification from human naïve pluripotent stem cells (PSCs). The factors regulating human TE specification remain poorly understood, but recently developed methods for deriving TE and trophoblast stem cells (TSCs) from naïve PSCs offer new model systems to gain insights into mechanisms of early placental development. The authors use human naïve PSCs generated with their recently developed 4CL culture medium to produce TE and TSCs. Based on shRNA knockdowns, they show that VGLL1 is required to generate TE from human naïve PSCs. They also map the genome-wide binding sites of VGLL1 by CUT&Tag analysis, which reveals enrichment of the TEAD4 DNA binding motif at VGLL1-bound sites. This leads them to hypothesize that VGLL1 interacts with TEAD4 to promote human TE specification. Indeed, TEAD1 and VGLL1 interact with each other based on reciprocal co-IP experiments. A role for VGLL1 in regulating H3K27ac deposition is inferred from a global reduction in H3K27ac levels upon VGLL1 knockdown and the strong enrichment of H3K27ac at VGLL1-TEAD4 co-bound sites. Finally, the authors show that VGLL1 is also required to sustain a trophoblast identity in expandable TSCs.

This is an interesting manuscript that sheds new light on the mechanisms regulating human TE specification. Strengths of this paper are the use of human naïve PSCs to investigate TE specification and the computational analyses of VGLL1 expression in human, monkey and mouse embryos. However, several additional experiments and analyses are required to better support the main conclusions.”

Response: We thank the reviewer for considering our work “interesting” and saying that it “sheds new light on the mechanisms regulating human TE specification”.

“Specific points:

1. The main conclusions rely on shRNAs to knock down VGLL1 expression in TE (Fig. 3 and 5) and TSCs (Fig. 6). While the hairpins achieve good knockdown efficiency, a general concern with this approach is the potential for off-target effects. Since VGLL1 is not expressed in pluripotent cells, it should be possible to knock out this gene in naïve PSCs and demonstrate

that TE specification is affected using a clean homozygous deletion.”

Response: We thank the reviewer for this insightful comment. As suggested, we have now generated *VGLLI* knockout (KO) clones (C45 and C68) with clean homozygous deletion using H9 human embryonic stem cells (ESCs) through CRISPR-Cas9 technology. Both *VGLLI* KO clones, compared to wild-type (WT) clones, showed reduced TE gene expression determined by RT-qPCR and RNA-seq as well as compromised cell proliferation confirmed by morphology and cell counting in TELC induction. These observations are consistent with the shRNA-mediated *VGLLI* knockdown (KD) findings. We have added these data in the revised manuscript (NEW Fig. 3i-n; revised manuscript line 197-204 page 7).

“2. The authors use the recently reported “4CL” conditions for naïve PSCs (Mazid et al., 2022). These conditions were only published very recently and by the same lab. To ensure the broad relevance of this work, it would be helpful to reproduce at least some of the key findings using TE cells that were generated from naïve PSCs in more widely established culture conditions, such as PXGL or 5iLA.”

Response: We agree with the reviewer that reproducing key findings using TELCs generated with other naïve PSC culture conditions will further strengthen and broaden the applicability of our observations. To do this, we have generated naïve human ESCs using PXGL and HENSM, two well-known human naïve media^{1,2}, and transduced them with *VGLLI* and *TEAD4* shRNAs along with *shLuc* as control. We then performed TELC induction using these transduced cells. Both *VGLLI* and *TEAD4* KD cells, compared to the *shLuc* control, lacked typical TE-like colony morphology and displayed substantially reduced expression of TE genes in both PXGL and HENSM. We also noticed compromised cell proliferation in *VGLLI* and *TEAD4* KD cells compared to the *shLuc* control. These findings demonstrate a high degree of consistency with the TELCs produced using 4CL. These data have been included in the revised manuscript (NEW Supplementary Fig. 2d-i; revised manuscript line 205-212 page 7 and NEW Supplementary Fig. 3m-r; revised manuscript line 244-247 page 8).

“3. Related to the previous point, it would be helpful if the authors could demonstrate using bioinformatic integration that their d5 TE cells in fact cluster more closely with pre-implantation TE than post-implantation CTBs. They cite the work from Io et al. and Guo et al., but those studies used naïve PSCs derived under different conditions. In their prior study (Mazid et al.), the authors directly converted 4CL naïve cells into TSCs and did not generate pre-implantation TE cells.”

Response: We thank the reviewer for pointing this out. To assess the fidelity with which TELCs derived from 4CL naïve ESCs reflect preimplantation embryo TE cells, we performed scRNA-seq for TELC-D5 cells and integrated them with scRNA-seq datasets of 4CL naïve ESCs³, human preimplantation^{4,5} and postimplantation embryo cells⁶. UMAP representation showed that TELC-D5 cells clustered tightly with human preimplantation embryo TE cells, but not with post-implantation cytotrophoblasts (CTBs). We have added these data in the revised manuscript (NEW Fig. 1e,f; revised manuscript line 129-135 page 5).

“4. In Fig. 2a-d the human embryo data from Petropoulos *et al.* (2016) are used to examine the expression of *VGLL1* and other TE regulators. This is a good starting point, but I think it would be instructive to look at another human embryo single cell dataset as well, for example Meistermann *et al.*, *Cell Reports*, 2021. A potential concern about the Petropoulos data is that a subset of their EPI and PE cells likely originate from TE (Stirparo *et al.*, 2018).”

Response: We are grateful to the reviewer for the very thorough scrutiny of the data. Following the reviewer’s suggestion, we have now performed the same type of analysis as in previous Fig. 2a-d using human embryo scRNA-seq dataset from Meistermann *et al.*⁵. UMAP analysis identified morula, pre-lineage, TE, ICM, epiblast (EPI) and primitive endoderm (PE) cells. TE genes such as *TFAP2C*, *TEAD4*, *GATA3* and *YAPI* were upregulated in human TE cells compared to pre-lineage and ICM cells, while *VGLL1* was exclusively expressed in TE cells. The pseudotime analysis using this dataset revealed that both TE and ICM lineages were originated from pre-lineage cells and, notably, *VGLL1* displayed dramatic upregulation at the onset of the TE branch. Therefore, the results utilizing the dataset from Meistermann *et al.* are in line with the results from Petropoulos *et al.* We have added these data in the revised manuscript (NEW Supplementary Fig. 1g-j; revised manuscript line 168-170 page 6).

“5. In Fig. 3C and Fig. 6a it would be helpful to use higher magnification images to see the impact of *VGLL1* knockdown on TE and TSC morphology.”

Response: We thank the reviewer for pointing this out. As suggested, we have now substituted previous images of these panels for higher magnification ones in the revised manuscript (NEW Fig. 3c and NEW Fig. 6a).

“6. Motif analysis at *VGLL1* binding sites in d5 TE reveals enrichment of the *TEAD4* motif (Fig. 4d). Is this analysis specific for *TEAD4* and not its paralogs, *TEAD1* and *TEAD3*? The latter two are also expressed in human trophoblast cells and may have overlapping functions.”

Response: Thanks for raising this question. At *VGLL1* binding sites in TELC-D5 cells, we also discovered *TEAD1* and *TEAD3* motifs. However, considering the important role of *TEAD4* in mouse TE lineage specification, we chose *TEAD4* rather than *TEAD1* or *TEAD3* to investigate its relationship with *VGLL1* in this study. This result has been added in the revised manuscript (NEW Fig. 4d).

“7. *TEAD4* and *VGLL1* interact with each other based on co-IP assays (Fig. 4f) and co-occupy target genes in d5 TE cells (Fig. 4h-i). Based on these data, the authors conclude that “cooperation between *VGLL1* and *TEAD4* is necessary for human TE induction from naïve PSCs” (line 223-224). Additional evidence is needed to support this assertion. For example, does *VGLL1* knockdown – or knockout - impair the ability of *TEAD4* to bind its target genes?”

Response: We appreciate the reviewer bringing up this issue. We previously demonstrated that *VGLL1* forms a strong partnership with *TEAD4* and both share a large proportion of their genomic binding sites. To further support our claim that the functional cooperation between *VGLL1* and *TEAD4* is crucial for the induction of human TELCs, we assessed *TEAD4* genomic binding using CUT&Tag in *VGLL1* KO cells at day

5 of TELC induction, as suggested by the reviewer. We noticed that the average genome binding intensity of TEAD4 was impaired in the *VGLL1* KO cells compared to WT control. This result further confirms our previous observation that cooperation between *VGLL1* and TEAD4 is required for inducing human TELCs from naïve PSCs. We have included these data in the revised manuscript (NEW Fig. 5j; revised manuscript line 322-327 page 11).

“8. Global levels of H3K27ac decline upon *VGLL1* knockdown (Fig. 5a) and H3K27ac is enriched at *VGLL1*-bound sites. This leads the authors to conclude that *VGLL1* is involved in opening chromatin at target loci through increased histone acetylation. It would be more convincing if the authors could include ATAC-seq and H3K27ac CUT&Tag analysis on *VGLL1* knockdown – or, ideally, knockout – cells, showing a specific depletion of open chromatin at *VGLL1* target sites.”

Response: We thank the reviewer for bringing up these very relevant points. As suggested, we have now performed H3K27ac CUT&Tag and Assay for Transposase Accessible Chromatin with high-throughput sequencing (ATAC-seq) for WT and *VGLL1* KO cells in TELC induction at day 5. We noticed that in *VGLL1* KO cells compared with WT control, the average enrichment of H3K27ac at *VGLL1* binding sites was reduced and the chromatin accessibility at these regions was also lower. These findings further reinforce our previous experiments indicating that *VGLL1* is involved in opening chromatin at target loci through increasing histone acetylation. We have included these data in the revised manuscript (NEW Fig. 5k, l; revised manuscript line 322-328 page 11).

“9. Fig. 6 shows that *VGLL1* is required not only in pre-implantation TE, but also in expandable TSCs that resemble post-implantation CTBs. Since TSCs derived from naïve PSCs have some different epigenetic properties from primary TSCs, can the authors reproduce these findings using a TSC line that was derived directly from blastocysts or placental tissues?”

Response: We thank the reviewer for this suggestion, as this will broaden the implications of our observation in this study. We have carried out the *VGLL1* KD experiment using a blastocyst-derived TSC line (TSC^{BT})⁷, as suggested by the reviewer. *VGLL1* KD in this cell line downregulated TSC gene expression as well as impaired cell proliferation, which is consistent with the findings using TSCs generated from human PSC-derived TELC-D5 cells. We have added these data in the revised manuscript (NEW Supplementary Fig. 6j-l; revised manuscript line 350-352 page 12).

“10. In Fig. 7 the authors integrate the CUT&Tag data for *VGLL1* and TEAD4 with previously published ChIP-seq data for GATA3 and TFAP2C to argue that all four factors cooperate in the human TE regulatory network. Aside from technical differences related to the comparison of CUT&Tag vs. ChIP-seq data, the GATA3 and TFAP2C data were generated using primed human PSCs treated with BMP4 (Krendl et al., 2017). As the authors themselves point out, this model does not recapitulate TE differentiation (line 367). Therefore, I’m confused why they rely on these previously published ChIP-seq data from the BMP4 model. It would be better to perform CUT&Tag for GATA3 and TFAP2C in the same d5 naïve TE cells that are analyzed throughout this manuscript.”

Response: Thanks for pointing this out. To better understand the interplay between TEAD4, VGLL1, GATA3 and TFAP2C in human TE gene regulatory network, we have now performed GATA3 and TFAP2C CUT&Tag with H9 4CL ESC-derived TELC-D5 cells. In line with prior findings, GATA3 and TFAP2C both displayed significant ratios of co-binding with VGLL1 and TEAD4, indicating a cooperative functional relationship between them. Notably, using the new datasets, we found that VGLL1 and TEAD4 shared around 70% of their binding sites with GATA3 (79.47% and 69.62% respectively), and less than 50% with TFAP2C (49.12% and 45.52% respectively), suggesting a stronger functional correlation between VGLL1, TEAD4 and GATA3. A relatively lower (37.98%) co-binding between GATA3 and TFAP2C was observed. In a four-way Venn diagram, VGLL1-TEAD4-GATA3-TFAP2C and VGLL1-TEAD4-GATA3 had the most substantial overlaps, whereas VGLL1-TEAD4-TFAP2C had a much lower overlap. Further investigation of these two types of regions revealed that genes co-bound by all four factors are mainly related with placenta development, cell growth regulation and histone modification, whereas genes co-bound by VGLL1-TEAD4-GATA3 are associated with multiple lineage development, suggesting a diverse functional outcome when the co-binding partners are altered. Besides, TEAD4 and TFAP2C motifs were enriched in GATA3 binding loci, and TEAD4 and GATA3 motifs were enriched in TFAP2C binding loci. Notably, VGLL1, TEAD4, GATA3 and TFAP2C co-bound to cell cycle and TE identity genes including their own regulatory regions, indicating the existence of positive feedback mechanisms. These findings are in consistent with our prior observations. We have replaced the original figures by these new data in the revised manuscript (NEW Fig. 7a-g; revised manuscript line 413-440 page 14-15).

Reviewer #2 (Remarks to the Author):

“In this work, Yueli Yang and colleagues sought to investigate the role of VGLL1 in early human trophectoderm (TE) development by employing recently developed model of naïve pluripotent cells to TE-like state transdifferentiation. They showed that VGLL1 is expressed at high level in human TE cells derived from naïve PSC, as well as in the TE compartment of human and monkey blastocysts. As reported before (PMID: 29361559), the authors demonstrated the interaction between VGLL1 and TEAD4, and its importance in early TE development. Using Cut and Tag, ATAC-seq, RNA-seq and ChIP-seq, they show that many regions occupied by VGLL1 occupied as well by TEAD4. By knocking down either VGLL1 or TEAD4 they show that the conversion to TE-like state is impaired mainly due to cell cycle arrest and reduced TE-specific gene expression. This notion was also true for TSC-derived from the same transdifferentiation model. Finally, they show that GATA3 and TFAP2C also occupy a large fraction of these regions but work independently. Overall, this is a very nice study, and the majority of the experiments and results are of high quality and convincing. However, some issue should be addressed prior to publication.”

Response: We thank the reviewer for the positive comments and considering our data as “of high quality” and “convincing”.

“Major comments:

1. One main concern of the reviewer is the claim regarding the importance of *VGLL1* in inducing the TE-like state. The reviewer is convinced that the cell cycle is impaired in the absence of *VGLL1* or *TEAD4* but is concerned whether the entire effect seen in the knockdown (KD) cells is related to cell cycle solely. First, the reduction of TE-specific genes is relatively mild in the KD cells (*CGB* and *CGA* are not TE-specific markers and are expressed in differentiated cells as well) and the morphology of the KD cells seems similar to the reviewer as to the control, albeit the colonies are definitely smaller. To exclude the possibility that the effect seen is only due to cell cycle arrest, the reviewer suggests to KO/KD *p53* in the naïve pluripotency *VGLL1* and *TEAD4* KD cells and repeat the experiment. Showing normal proliferating cells and reduced expression of TE-specific genes will validate the authors’ claim.”

Response: We appreciate the reviewer for bringing up this issue. Following the reviewer’s suggestion, we have generated two effective shRNAs targeting *TP53* (*shTP53-1* and *shTP53-2*). We then transduced 4CL H9 WT ESCs, two *VGLL1* KO clones (C45 and C68) and two *TEAD4* KO clones (C132 and C171) with lentiviruses for these two shRNAs and used *shLuc* as control. These cells were then used for TELC induction. In all settings, we observed a 50-70% increase of cell number in *TP53* knockdown (KD) cells relative to control cells, as expected. Despite the increased cell number, we did not observe upregulation of TE genes like *GATA2*, *GATA3*, *TFAP2C* and *YAPI* in *TP53* KD cells compared with control cells. These results provide evidence that the regulatory function of *VGLL1* and *TEAD4* on TE gene expression is not due to cell cycle arrest. We have added these results in the revised manuscript (NEW Supplementary Fig. 2j-l; revised manuscript line 213-221 page 7-8 and NEW Supplementary Fig. 4a-c; revised manuscript line 247-248 page 8-9).

“2. Is there a reason why the authors used shRNAs and not CRISPR/Cas9 mediated KO? ShRNAs known to have non-specific effect especially on proliferation so if possible, the authors should repeat at least one main assay with KO cells.”

Response: We agree with the reviewer that repeating at least one of the main assays with KO cells will enhance our current conclusions. As also suggested by reviewer 1 above, we have generated *VGLL1* knockout (KO) clones (C45 and C68) with clean homozygous deletion using H9 human ESCs through CRISPR-Cas9 technology. Consistent with the shRNA-mediated *VGLL1* KD, both *VGLL1* KO clones, compared to wild-type (WT) clones, showed reduced TE gene expression determined by RT-qPCR and RNA-seq, as well as compromised cell proliferation confirmed by morphology and cell counting. The downregulated genes in *VGLL1* KO cells were associated with *in utero* embryonic development, placenta development, regulation of cell growth and positive regulation of cell cycle, as determined by GO enrichment analysis. These findings further support that *VGLL1* is a key regulator of human TE lineage specification. We have added these data in the revised manuscript (NEW Fig. 3i-n; revised manuscript line 197-204 page 7).

Similarly, we have generated *TEAD4* KO clones (C131 and C172) using H9 ESCs. Both *TEAD4* KO clones showed reduced TE gene expression as well as compromised cell proliferation compared to WT clones in TELC conversion. The downregulated genes in

TEAD4 KO cells were involved in placenta development and cell cycle regulation. These findings with TEAD4 KO are also consistent with the observations in shRNA-mediated TEAD4 KD cells. We have added these data in the revised manuscript (NEW Supplementary Fig. 3f-i; revised manuscript line 244-247 page 8).

“3. To support the authors’ claim that VGLL1 and TEAD KD cells are indeed impaired, a functional assay should be conducted. This can be shown for example by directed differentiation toward STs and EVTs.”

Response: This is a very good point. To assess the functional role of VGLL1 and TEAD4 in TSC to SCT and EVT differentiation, expandable TSCs derived from TELC-D5 cells generated from 4CL H9 ESCs were transduced with two effective shRNAs for each gene along with shLuc as control. Subsequently, SCT and EVT differentiation were carried out as previously described⁸. The shLuc control cells both in SCT or EVT induction displayed normal cell proliferation and a distinct morphological change, but both VGLL1 and TEAD4 KD cells ceased proliferating from the beginning of the induction process and cell death was gradually increased until all cells died around day 5, demonstrating the importance of both VGLL1 and TEAD4 in maintaining the functional potential of TSCs. We have added these data in the revised manuscript (NEW Supplementary Fig. 6m, n; revised manuscript line 352-358 page 12 and NEW Supplementary Fig. 7v, w; revised manuscript line 369-370 page 12).

“4. The authors show that TE-like and TSC VGLL1 KD cells still express pluripotency genes (Figs. 3e and 6c). While it could be reasonable for the TE-like cells the reviewer is confused as to how this can happen in TSCs? Are the authors claim that VGLL1 constantly suppresses pluripotency genes at the TSC stage? The authors should show the expression of some of those pluripotency genes at the protein level (staining or western blot) and explain this result.”

Response: We thank the reviewer for this insightful comment. To assess pluripotency genes expression at protein level, we have performed Western blotting for OCT4, NANOG and SOX2 in TELC-D5 cells and TSCs derived from 4CL H9 ESCs, and found that these genes actually do not express at protein level after 4CL ESCs differentiate into TELCs or TSCs. In our revised manuscript, we have included these Western blotting results and described more clearly this part (NEW Fig. 1i; revised manuscript line 144-145 page 5, revised manuscript line 191-192 page 7 and revised manuscript line 339-342 page 11-12). We hope that the reviewer will be satisfied with our modification but will be happy to add more analyses, experiments or discussion if s/he considers it necessary.

“5. Given the high overlap between GATA3 and TFAP2C to VGLL1 regions, the reviewer believe it is important to show whether they are working as a complex by performing CO-IP experiments.”

Response: We thank reviewer for raising this issue. We have now conducted reciprocal co-immunoprecipitation (co-IP) for VGLL1, GATA3 and TFAP2C in TELC-D5 cells. When we pull-down VGLL1, both GATA3 and TFAP2C can be detected in the IP sample. We can also detect VGLL1 and TFAP2C in the GATA3 pull-down IP sample, and VGLL1 and GATA3 in TFAP2C pull-down IP sample. Besides, we performed new

CUT&Tag for GATA3 and TFAP2C in TELC-D5 cells, as suggested by Reviewer 1 (item 10), and further validated the cooperation between VGLL1, GATA3 and TFAP2C from a genomic binding perspective (described in detail in the response to reviewer 1, item 10). These results strongly support that these three factors can work as a complex in human TELC differentiation from naïve PSCs. We have added these data in the revised manuscript (NEW Supplementary Fig. 8m; revised manuscript line 432-433 page 14-15).

“6. It is not clear how the authors explain the results shown in Fig. 6j? The regions (group 2) that separate the role of VGLL1 in TE vs TSCs are not relevant to the TE state at all. How do the authors explain enrichment for muscle contraction, blood circulation and gland and eye development in these cells? Could it be that these results actually demonstrate a non-homogenous differentiation toward the TSC state at day 5 of differentiation? Could it be that TE-like cells in this protocol contain also other differentiated cells? from different cell types?”

Response: We thank the reviewer for pointing this out. Group 2 genes, which are bound by VGLL1 and TEAD4 only in TELC-D5 cells, are involved in various development-related biological functions. Some of these functions were group 2 specific like muscle contraction and blood circulation, while others, crucially, and some were shared with group 1 and/or group 3 functions, like the establishment or maintenance of cell polarity, *in utero* embryonic development and placenta development, which are relevant to the TE state (see upper part of the PREVIOUS and NEW Fig. 6i for group 2 GO terms). Thus, group 2 genes indeed play crucial roles in TELC differentiation. We speculate the pan-developmental-related binding of VGLL1 and TEAD4 in group 2 genes correlates with an unstable preliminary extraembryonic cell fate, in which the chromatin regions of different lineage genes were still accessible for binding. In this regard, TE cells isolated from human blastocysts have been shown to be able to integrate into the ICM and begin to express NANOG, suggesting that human TE cells are not yet committed⁹. We have now clarified this point in the text (revised manuscript line 387-393 page 13).

Regarding the homogeneity, we have performed scRNA-seq of TELC-D5 cells and integrated these cells with the starting naïve ESCs, preimplantation human embryo and postimplantation human embryo cells. TELC-D5 cells clustered tightly with human preimplantation embryo TE cells and their gene expression was comparable (NEW Fig. 1e, f; revised manuscript line 129-135 page 5). Pearson correlation analysis among the subclusters within TELC-D5 cells showed that they are highly correlated (correlation score > 0.9) except for two minor subclusters (cluster 7 and 8), which only account for 2.08 % of TELC-D5 cells. Cluster 7 cells expressed SCT related genes. Additionally, we examined the expression of marker genes for other lineages in TELC-D5 cells but were unable to detect their expression. These results indicate that 4CL naïve ESC-derived TELC-D5 cells are relatively homogenous and resemble the human preimplantation embryo TE. We have added the new data and explanations in the revised manuscript (NEW Supplementary Fig. 1b-d; revised manuscript line 135-140 page 5).

“Minor comments:

7. Fig. 1c- the reviewer think it is important to show that the examined genes are not expressed in primed ESCs. Please include primed ESCs to the staining panel.”

Response: Thanks for the comment. We have performed immunostaining for primed ESCs and included the images in the revised manuscript (NEW Supplementary Fig. 1a).

“8. Since no study rigorously demonstrated that following 5 days of differentiation the cells are bona fide TE cells, the authors should change the term “TE” to “TE-like” throughout the entire manuscript.”

Response: This is a good point and we have changed the term ‘TE’ (trophectoderm) to ‘TELC’ (trophectoderm-like cell) throughout the revised manuscript.

“9. Fig. 2d, 2h- Title is missing for the Y axis.”

Response: Thanks for pointing out this. We have now added the Y axis title.

“10. Fig. 3f- more TE/TSC-specific genes should be examined such as GATA2, TFAP2C, TFAP2A, KLF5 etc.”

Response: Thanks for the suggestion. We have now included RT-qPCR for these genes into NEW Fig. 3f.

“11. The authors conducted ATAC-seq and examined loci co-bound by VGLL1 and TEAD. However, they did not analyze loci bound only by VGLL1 or TEAD4. Can this be added to the manuscript.”

Response: We thank the reviewer for bringing up this question. To do this, we have analyzed the overlap among VGLL1 binding sites, TEAD4 binding sites and the open chromatin regions in TELC-D5 cells. Most open chromatin regions were co-bound by VGLL1 and TEAD4 and nested genes were related with functions like placenta development and cell cycle regulation. On the contrary, the number of open chromatin loci bound only by VGLL1 or TEAD4 were much less, and primarily involved in other system development or structure organization. We have added these data in the revised manuscript (NEW Supplementary Fig. 5d-f; revised manuscript line 318-320 page 11).

“12. It has been suggested lately that GATA3 binds to VGLL1 promoter in trophoblast progenitor and increases its expression compared to naïve ESCs (PMID: 35637409). Here, the authors claim that VGLL1 is also a regulator of GATA3 at the same developmental stage. Can the authors test GATA3 KD transdifferentiation experiment and compare it to VGLL1 KD differentiation experiment, in at least one main assay.”

Response: We thank the reviewer for this suggestion. We have performed GATA3 KD during TELC induction from 4CL H9 ESCs. Similar to VGLL1, GATA3 KD results in downregulation of TE gene expression. We have added these data in the revised manuscript (NEW Supplementary Fig. 8k, l; revised manuscript line 420-422 page 14).

Reviewer #3 (Remarks to the Author):

“In this report, the authors define a role for VGLL1 as a co-factor for TEAD4 in the regulation of the trophoctodermal (TE) and trophoblast stem (TS) cell states. The authors provide a

breadth of experimental work to determine expression profiles and loss-of-function experiments on the TE and TS cell states. In addition, the authors examine genome-wide distributions of VGLL1 and TEAD4 in order to identify direct targets of TEAD4 and VGLL1 action. The role of VGLL1 as a regulator of TEAD4 action was contrasted to roles for YAP and WWTR1 (TAZ) in TEAD4 action. Furthermore, species specific biology of TE and TS cell stem states were investigated. The manuscript is well-written and addresses an important biological problem. The work represents a significant contribution to understanding regulatory mechanisms controlling trophoblast lineage development; however, some issues with the experimentation and interpretation of data are evident. Specific concerns are elaborated below.”

Response: We thank the reviewer for the careful assessment and for considering our manuscript is “well-written”, “important” and “represents a significant contribution”.

“1) The experimentation was performed with the H9 human embryonic stem (ES) cell line. These cells were used to create naïve pluripotent stem cells and then these naïve pluripotent stem cells were converted to trophoctoderm (TE) cells and to trophoblast stem (TS) cells. The role of VGLL1 was investigated in these cell models derived from H9 human ES cells. Not all pluripotent stem cells exhibit similar abilities to differentiate into trophoblast cell lineages (Cinkornpumin et al. Stem Cell Reports. 2020;15(1):198-213, PMC7363941). Thus, it is important to demonstrate that the phenomena and mechanisms described in this report are not unique to derivatives of H9 ES cells. Do H9 ES cell-derived TS cells possess the capacity to differentiate into extravillous trophoblast and syncytiotrophoblast? Utilization of cytotrophoblast-derived TS cells as a reference standard for some of the work would be very helpful.”

Response: We thank the reviewer for these insightful comments. We will answer these questions in four parts: 1) demonstrate the phenomenon with another cell line; 2) demonstrate the mechanism with another cell line; 3) check VGLL1 and TEAD4 function in TSC differentiation to SCT and EVT; 4) repeat some of the work with embryo-derived TSCs.

1) We fully agree that trophoblast lineage differentiation potential may vary among cell lines and this could impact the specific phenotypes. Therefore, to exclude this possibility and strengthen our conclusions, we performed VGLL1 and TEAD4 knockdown (KD) in another human PSC line: UH10 iPSCs^{10,11}. We induced UH10 iPSCs into a 4CL naïve state and knocked down VGLL1 and TEAD4 with two effective shRNAs for each gene during both TELC-induction process and TSC maintenance. Both VGLL1 and TEAD4 KD resulted in reduced TE/TSC gene expression determined by RT-qPCR, as well as compromised cell proliferation confirmed by morphology and cell counting (NEW Supplementary Fig. 2a-c; revised manuscript line 205-212 page 7, NEW Supplementary Fig. 3j-l; revised manuscript line 244-247 page 8, NEW Supplementary Fig. 6g-i; revised manuscript line 350-352 page 12 and NEW Supplementary Fig. 7p-r; revised manuscript line 367-369 page 12). These findings demonstrate that VGLL1 and TEAD4 are required for both the induction of human TELCs and the maintenance of TSC identity, regardless of the cell line used.

2) To demonstrate that the mechanism is not only specific to H9 ESCs, we performed

reciprocal co-immunoprecipitation for *VGLL1* and *TEAD4* in 4CL UH10 iPSC-derived TELC-D5 cells and TELC-derived TSCs and found that *TEAD4* is strongly bound to *VGLL1* (NEW Supplementary Fig. 3a; revised manuscript line 241-244 page 8 and NEW Supplementary Fig. 7a; revised manuscript line 359-364 page 12). Next, we examined the histone acetylation levels after *VGLL1* KD in 4CL UH10 iPSC-derived TELC-D5 cells and TSCs by Western blotting. We observed that histone H3 acetylation (H3ac) and H3K27ac levels were decreased in TELC-D5 cells and TSCs following *VGLL1* KD, which is consistent with the findings using 4CL H9 ESC-derived TELCs and TSCs (NEW Supplementary Fig. 5a; revised manuscript line 287-293 page 10 and NEW Supplementary Fig. 8c; revised manuscript line 374-376 page 13). These results demonstrate that the mechanisms we report in this study are not unique to H9 ESCs but are more broadly applicable to other cell lines as well.

3) We and others have demonstrated that H9 ESC-derived TSCs are capable of differentiating into SCT and EVT^{3, 12, 13}. To assess the functional role of *VGLL1* and *TEAD4* during TSC differentiation to SCT and EVT, TSCs derived from TELC-D5 cells generated from 4CL H9 ESCs were transduced with two effective shRNAs for each gene along with *shLuc* as control. SCT and EVT differentiation were carried out as previously described⁸. The *shLuc* control cells both in SCT or EVT induction displayed normal cell proliferation and a distinct morphological change, but *VGLL1* and *TEAD4* KD cells ceased proliferating from the beginning of the induction process and cell death was gradually increased until all cells died around day 5 (NEW Supplementary Fig. 6m, n; revised manuscript line 352-358 page 12 and NEW Supplementary Fig. 7v, w; revised manuscript line 369-370 page 12), demonstrating the importance of both *VGLL1* and *TEAD4* in maintaining the functional potential of TSCs.

4) To further validate our results with human embryo-derived TSCs as reference, which is also suggested by reviewer 1 (item 9), we also carried out the *VGLL1* and *TEAD4* KD experiment using a blastocyst-derived TSC line (TSC^{BT})⁷. Both *VGLL1* and *TEAD4* KD resulted in downregulation of TSC gene expression as well as impaired TSC proliferation which is consistent with the findings using TSCs generated from human PSC-derived TELCs. We have added these data in the revised manuscript (NEW Supplementary Fig. 6j-l; revised manuscript line 350-352 page 12 and NEW Supplementary Fig. 7s-u; revised manuscript line 367-369 page 12).

“2) The findings published in this report emphasize the importance of *VGLL1* in serving as a cofactor for *TEAD* action in human *TE* and *TS* cells and suggest that *YAP1* or *WWTR1* are not serving as *TEAD* cofactors in these cell lineages. There are published reports implicating *YAP1* and *WWTR1* as cofactors for *TEAD* family transcription factors in maintenance of the human *TS* cell/cytotrophoblast stem state (Meinhardt et al. *Proc Natl Acad Sci U S A.* 2020; 117(24):13562-13570. PMC7306800; Saha et al. *Proc Natl Acad Sci U S A.* 2020; 117(30):17864-17875. PMC7395512; Ray et al. *Proc Natl Acad Sci U S A.* 2022; 119(36):e2204069119. PMC9457323). These reports directly utilized trophoblast cell populations isolated from placenta and verified aspects of their work with trophoblast tissues. Is there a difference in roles for *VGLL1*, *YAP1*, and *WWTR1* as *TEAD* cofactors based on the *in vitro* model system utilized? Are the authors results unique to *TE* and *TS* cells derived from

H9 ES cells? Are actions for YAP1 and WWTR1 in TS cells published by others compatible with the authors observations? The authors need to address these issues.”

Response: We thank the reviewer for raising these important questions. In our previous manuscript, we demonstrated that TEAD4 interacts strongly with VGLL1 and displays rather less interaction with YAP (PREVIOUS Fig. 4f, Fig. 6e and manuscript lines 206-210, 287-289). However, we did not test the interaction of TEAD4 with WWTR1. To better elucidate the functional relationship of different co-factors with TEAD4 in human, we performed TEAD4 co-immunoprecipitation in TELC-D5 cells and TSCs derived from 4CL H9 ESCs and 4CL UH10 iPSCs to exclude the potential cell line-specific phenomenon. Consistent with the published reports, we observed in TELCs and TSCs derived from both cell lines that TEAD4 interacts with YAP and WWTR1, though to a lesser extent than with VGLL1 (NEW Fig. 4e, NEW Supplementary Fig. 3a; revised manuscript line 251-255 page 9 and NEW Fig. 6e, NEW Supplementary Fig. 7a; revised manuscript line 359-364 page 12). We also found for both cell lines that the KD of *YAP1* or *WWTR1* reduced TELC induction as well as impaired TSC maintenance (NEW Supplementary Fig. 4d-k; revised manuscript line 255-258 page 9 and NEW Supplementary Fig. 7b-i; revised manuscript line 361-364 page 12). These results demonstrate that the interaction between YAP or WWTR1 with TEAD4 and their necessity is not specific to trophoblast cell populations isolated from placenta. Additionally, these observations were not unique to TELCs and TSCs derived from H9 ESCs.

To further illuminate how VGLL1 works in tandem with TEAD4 along with other co-factors like YAP to regulate human TELC induction, we performed CUT&Tag for YAP at day 5 of TELC induction with H9 4CL naïve ESCs. Co-occupancy analysis between VGLL1, TEAD4 and YAP revealed that VGLL1 and TEAD4 share a substantial number (43,048) of genomic binding regions, among which 57 % regions were also co-bound by YAP (24,626), including cell cycle/self-renewal genes (NEW Fig. 4g-i; revised manuscript line 259-274 page 9). These data are consistent with the co-immunoprecipitation results showing that TEAD4 interacts more strongly with VGLL1 relative to YAP. In addition, subcellular fractionation followed by Western blotting for TELC induction at day 5 showed that VGLL1 and TEAD4 mainly localized in the nucleus (NEW Supplementary 4l; revised manuscript line 274-278 page 9), while YAP was more restricted to the cytoplasm. The distinct subcellular distribution further supports the notion that TEAD4 display less interaction with YAP compared with VGLL1.

As a summary, TEAD4 recruits all three co-factors, with VGLL1 as the strongest partner, to regulate human TELC induction, regardless of the cell line used. All these three co-factors contribute to human TE lineage specification through regulating TE/TSC gene expression and self-renewal, which is compatible with reports using other *in vitro* models¹⁴. Please also see our response to the following question regarding the advantage of using different co-factors in human. We have added these data and described clearer in the revised manuscript.

“3) *The authors should consider discussing the evolutionary significance of utilizing different TEAD cofactors in mouse versus primate trophoblast cells. Is there an advantages in using*

distinct cofactors for mouse versus primate trophoblast cells? Does each cofactor direct TEAD action to unique parts of the mouse versus primate genome? ”

Response: We thank the reviewer for raising this point. There is substantial evidence to suggest that the human and mouse have considerable spatiotemporal differences regarding trophoblast lineage specification. It is an intriguing fact that although the use of TEAD4’s co-factors in mouse versus human TE specifications appears to be different, the ones in mice also play functional role in human. In mouse, TEAD4 mainly interacts with YAP/WWTR1 to regulate TE specification, while in human TEAD4 showed stronger interaction with VGLL1 compared with YAP or WWTR1. One possible explanation could be interspecies differences in the regulation of the Hippo pathway in early development. In human, YAP localized in the nucleus of both TE and ICM cells, and its activity promotes human naïve PSC generation¹⁵. Therefore, YAP could regulate ICM and TE bipotency rather than only lead to TE cell fate as in mouse. However, VGLL1 is highly and specifically expressed at the onset of human TE specification but not in mouse. From an evolutionary perspective, the interaction of both VGLL1 and YAP may provide more flexibility and further safeguard the extraembryonic trophoblast development in human (revised manuscript line 279-281 page 10).

Besides, by analysing CUT&Tag data for TEAD4, VGLL1 and YAP at day 5 of TELC induction, we found that VGLL1-YAP-TEAD4 co-bound to large number of genomic binding regions, and VGLL1 and YAP can individually co-bound with TEAD4 (NEW Fig. 4h; revised manuscript line 259-268 page 9). We performed GO analysis for genes co-bound by VGLL1, YAP and TEAD4, and those bound by either two of them in TELC-D5 cells. VGLL1-TEAD4-YAP co-bound or VGLL1-TEAD4-only genes were enriched in terms related to placenta development, cell cycle regulation and, interestingly, histone acetylation, while YAP-TEAD4-only co-bound genes were enriched for other biological processes such as protein polyubiquitination and positive regulation of autophagy. The distribution of these three factors across the genome supports the notion that VGLL1 and YAP have complementary functions by guiding TEAD4 to the common/same genomic regions (VGLL1-YAP-TEAD4 co-bound), and while serving distinct functions by directing TEAD4 to unique regions of the genome (VGLL1-TEAD4-only or YAP-TEAD4-only). We have added these data in the revised manuscript (NEW Fig. 4i; revised manuscript line 268-274 page 9).

“4) In the last set of data discussed in the Results section, the authors compare their findings with datasets generated from Krendl et al. 2017. These datasets from Krendl et al. were generated from BMP4-treated human pluripotent stem cells. As stated in the report, this population of cells exhibits similarities to amnion and not trophoblast. Thus, the merits of using these BMP4-treated human pluripotent stem cell datasets need to be better justified. Reliance exclusively on the expression of APA in a cell population could be misleading.”

Response: This is a good comment and this point was similarly raised by reviewer 1 (item 10). To better understand the interplay between TEAD4, VGLL1, GATA3 and TFAP2C in orchestrating the human TE regulatory network, we have now performed GATA3 and TFAP2C CUT&Tag with 4CL ESC-derived TELC-D5 cells. In line with the prior findings, GATA3 and TFAP2C both displayed significant ratios of co-binding with

VGLL1 and TEAD4, indicating a cooperative functional relationship between them. Notably, using the new datasets, we found that VGLL1 and TEAD4 shared around 70% of their binding sites with GATA3 (79.47% and 69.62% respectively), and less than 50% with TFAP2C (49.12% and 45.52% respectively), suggesting a stronger functional correlation between VGLL1, TEAD4 and GATA3. Additionally, a relatively lower (37.98%) co-binding between GATA3 and TFAP2C were observed. In a four-way Venn diagram, the VGLL1-TEAD4-GATA3-TFAP2C and VGLL1-TEAD4-GATA3 had the most substantial overlaps, whereas VGLL1-TEAD4-TFAP2C had a much smaller overlap. Further investigation of these two types of regions revealed that genes co-bound by all four factors were mainly related with placenta development, cell growth regulation and histone modification, whereas genes co-bound by VGLL1-TEAD4-GATA3 were associated with multiple lineage development, suggesting a diverse functional outcome when the co-binding partners were altered. Besides, TEAD4 and TFAP2C motifs were enriched in GATA3 binding loci, and TEAD4 and GATA3 motifs were enriched in TFAP2C binding loci. Notably, VGLL1, TEAD4, GATA3 and TFAP2C co-bound to genes related to TE/TSC identity and cell cycle including their own regulatory regions, indicating the existence of positive feedback mechanisms. These findings are consistent with our prior observations in the original manuscript. We have replaced the original figures by these new data in the revised manuscript (NEW Fig. 7a-g; revised manuscript line 413-440 page 14-15).

References

- 1 Yanagida, A., et al. Naive stem cell blastocyst model captures human embryo lineage segregation. *Cell Stem Cell* **28**, 1016-1022 e4 (2021)
- 2 Bayerl, J., et al. Principles of signaling pathway modulation for enhancing human naive pluripotency induction. *Cell Stem Cell* **28**, 1549-1565.e12 (2021)
- 3 Mazid, M. A., et al. Rolling back human pluripotent stem cells to an eight-cell embryo-like stage. *Nature* **605**, 315-324 (2022)
- 4 Petropoulos, S., et al. Single-Cell RNA-Seq Reveals Lineage and X Chromosome Dynamics in Human Preimplantation Embryos. *Cell* **165**, 1012-26 (2016)
- 5 Meistermann, D., et al. Integrated pseudotime analysis of human pre-implantation embryo single-cell transcriptomes reveals the dynamics of lineage specification. *Cell Stem Cell* **28**, 1625-1640 e6 (2021)
- 6 Xiang, L., et al. A developmental landscape of 3D-cultured human pre-gastrulation embryos. *Nature* **577**, 537-542 (2020)
- 7 Okae, H., et al. Derivation of Human Trophoblast Stem Cells. *Cell Stem Cell* **22**, 50-63 e6

(2018)

- 8 Fan, Y., et al. Generation of human blastocyst-like structures from pluripotent stem cells. *Cell Discov* **7**, 81 (2021)
- 9 De Paepe, C., et al. Human trophectoderm cells are not yet committed. *Human Reproduction* **28**, 740-749 (2012)
- 10 Huang, K., et al. BMI1 enables interspecies chimerism with human pluripotent stem cells. *Nature Communications* **9**, (2018)
- 11 Zhu, Y., et al. Generating functional cells through enhanced interspecies chimerism with human pluripotent stem cells. *Stem Cell Reports* **17**, 1059-1069 (2022)
- 12 Io, S., et al. Capturing human trophoblast development with naive pluripotent stem cells in vitro. *Cell Stem Cell* **28**, 1023-1039 e13 (2021)
- 13 Soncin, F., et al. Derivation of functional trophoblast stem cells from primed human pluripotent stem cells. *Stem Cell Reports* **17**, 1303-1317 (2022)
- 14 Soncin, F., et al. Comparative analysis of mouse and human placentae across gestation reveals species-specific regulators of placental development. *Development* **145**, (2018)
- 15 Qin, H., et al. YAP Induces Human Naive Pluripotency. *Cell Rep* **14**, 2301-12 (2016)

REVIEWER COMMENTS

Reviewer #1 (Remarks to the Author):

The authors have substantially revised their manuscript in response to the points raised by me and the other reviewers and addressed all my concerns. They have now convincingly demonstrated that VGLL1 partners with TEAD4 to regulate human TE specification from naïve PSCs.

My only remaining suggestion is to replace the term “human PSC-derived synthetic embryos” (p. 17, line 495) with “human PSC-derived embryo models”, in accordance with the recent recommendation to avoid the term “synthetic embryo” (Landecker and Clark, Cell Stem Cell, 2023). I also advise updating the citations on line 495 to include the latest human blastoid and post-implantation embryo model papers that were published over the summer.

Reviewer #2 (Remarks to the Author):

The authors have undeniably invested significant time and effort into substantiating their claims, resulting in overall satisfaction from the reviewer. However, one critical issue must be addressed before publication.

The figure illustrating direct differentiation into EVT and ST cells lacks conviction. Firstly, the morphology of the ST cells appears incorrect and resembles EVT cells. Secondly, no markers have been presented to substantiate the differentiation claim. To enhance the credibility of this differentiation, the authors should consider staining the cells with EVT and ST markers, such as HLA-G for EVT and SDC1 for ST cells. This additional evidence will strengthen their findings and ensure the robustness of their conclusions.

Reviewer #3 (Remarks to the Author):

The authors have satisfactorily addressed my concerns.

Response to reviewers' comments

We thank all reviewers for their positive feedback and have now incorporated all remaining minor requests in the revised manuscript. A point-by-point response to the reviewers' comments is provided below and changes in the manuscript to address these comments are highlighted in red.

Reviewer #1 (Remarks to the Author):

“The authors have substantially revised their manuscript in response to the points raised by me and the other reviewers and addressed all my concerns. They have now convincingly demonstrated that VGLL1 partners with TEAD4 to regulate human TE specification from naïve PSCs.

My only remaining suggestion is to replace the term “human PSC-derived synthetic embryos” (p. 17, line 495) with “human PSC-derived embryo models”, in accordance with the recent recommendation to avoid the term “synthetic embryo” (Landecker and Clark, Cell Stem Cell, 2023). I also advise updating the citations on line 495 to include the latest human blastoid and post-implantation embryo model papers that were published over the summer.”

Response: We thank the reviewer for considering that we have now convincingly demonstrated our claims. As suggested, we have replaced the term ‘human PSC-derived synthetic embryos’ with ‘human PSC-derived embryo models’ in our revised manuscript (page 17, line 497) and updated the citations to include the latest research papers on human blastoid and post-implantation embryo models (page 17, line 497).

Reviewer #2 (Remarks to the Author):

“The authors have undeniably invested significant time and effort into substantiating their claims,

resulting in overall satisfaction from the reviewer. However, one critical issue must be addressed before publication.

The figure illustrating direct differentiation into EVT and ST cells lacks conviction. Firstly, the morphology of the ST cells appears incorrect and resembles EVT cells. Secondly, no markers have been presented to substantiate the differentiation claim. To enhance the credibility of this differentiation, the authors should consider staining the cells with EVT and ST markers, such as HLA-G for EVT and SDC1 for ST cells. This additional evidence will strengthen their findings and ensure the robustness of their conclusions.”

Response: We thank the reviewer for acknowledging our efforts in substantiating our claims and expressing overall satisfaction with our work. Regarding the morphology, we would like to clarify that those photos were taken at day 5 of differentiation (see previous legends for Supplementary Fig. 6m, n and Supplementary Fig. 7v, w), which is less than half of the entire induction process (see previous Methods section for ‘EVT and SCT differentiation’, line 766-773). In addition, the previous images had a limited resolution. To address this issue, we have now replaced the previous phase contrast images with higher-resolution ones for day 5 of differentiation and also included the end time point images for EVT (day 12) and SCT (day 14) differentiation, where distinct morphologies can be observed (see NEW Supplementary Fig. 6m, n and Supplementary Fig. 7v, w). We have also provided a clearer explanation of this in the text (revised manuscript line 356-359, page 12). Importantly, in response to the reviewer’s suggestion to enhance the credibility of this differentiation experiment, we have now performed staining for HLA-G in EVT cells and SDC1 in SCT cells and included those results in our revised manuscript (see NEW Supplementary Fig. 6m, n and Supplementary Fig. 7v, w). We hope that the reviewer will be satisfied with these changes and thank him/her for this additional comment, as it has helped us present things clearer.

Reviewer #3 (Remarks to the Author):

“The authors have satisfactorily addressed my concerns.”

Response: We thank the reviewer for acknowledging that we have satisfactorily addressed his/her concerns.